# Endomembrane targeting of human OAS1 p46 augments antiviral activity

Frank W Soveg[1,2†], Johannes Schwerk[1,2†], Nandan S Gokhale[1,2],
Karen Cerosaletti[3], Julian R Smith[1,2], Erola Pairo-Castineira[4], Alison M Kell[1,2,5],
Adriana Forero[1,2,6], Shivam A Zaver[7], Katharina Esser-Nobis[1,2], Justin A Roby[1,2‡],
Tien-Ying Hsiang[1,2], Snehal Ozarkar[1,2], Jonathan M Clingan[1,2],
Eileen T McAnarney[8], Amy EL Stone[1,2,9], Uma Malhotra[10,11], Cate Speake[3],
Joseph Perez[12], Chiraag Balu[1,2], Eric J Allenspach[13], Jennifer L Hyde[6],
Vineet D Menachery[8], Saumendra N Sarkar[12], Joshua J Woodward[2,7],
Daniel B Stetson[1,2], John Kenneth Baillie[4,14], Jane H Buckner[3], Michael Gale Jr[1,2],
Ram Savan[1,2]*

[1]Department of Immunology, School of Medicine, University of Washington, Seattle, United States; [2]Center for Innate Immunity and Immune Disease, University of Washington, Seattle, United States; [3]Benaroya Research Institute at Virginia Mason, Seattle, United States; [4]Roslin Institute, University of Edinburgh, Edinburgh, United Kingdom; [5]Department of Molecular Genetics and Microbiology, School of Medicine, University of New Mexico, Albuquerque, United States; [6]Department of Microbial Infection and Immunity, College of Medicine, The Ohio State University, Columbus, United States; [7]Department of Microbiology, School of Medicine, University of Washington, Seattle, United States; [8]Department of Microbiology and Immunology, University of Texas Medical Center, Galveston, United States; [9]Department of Basic Sciences, College of Osteopathic Medicine, Touro University Nevada, Henderson, United States; [10]Department of Infectious Disease, Virginia Mason Medical Center, Seattle, United States; [11]Department of Medicine, Section of Infectious Diseases, University of Washington, Seattle, United States; [12]Cancer Virology Program, University of Pittsburgh Cancer Institute, University of Pittsburgh, Pittsburgh, United States; [13]Center for Immunity and Immunotherapies, Seattle Children's Research Institute, Seattle, United States; [14]MRC Human Genetics Unit, Institute of Genetics and Molecular Medicine, University of Edinburgh, Western General Hospital, Edinburgh, United Kingdom

*For correspondence:
savanram@uw.edu

†These authors contributed equally to this work

Present address: ‡School of Biomedical Sciences, Charles Stuart University, Wagga Wagga, Australia

**Competing interests:** The authors declare that no competing interests exist.

**Abstract** Many host RNA sensors are positioned in the cytosol to detect viral RNA during infection. However, most positive-strand RNA viruses replicate within a modified organelle co-opted from intracellular membranes of the endomembrane system, which shields viral products from cellular innate immune sensors. Targeting innate RNA sensors to the endomembrane system may enhance their ability to sense RNA generated by viruses that use these compartments for replication. Here, we reveal that an isoform of oligoadenylate synthetase 1, OAS1 p46, is prenylated and targeted to the endomembrane system. Membrane localization of OAS1 p46 confers enhanced access to viral replication sites and results in increased antiviral activity against a subset of RNA viruses including flaviviruses, picornaviruses, and SARS-CoV-2. Finally, our human genetic analysis shows that the *OAS1* splice-site SNP responsible for production of the OAS1 p46 isoform correlates with protection from severe COVID-19. This study highlights the importance of endomembrane targeting for the antiviral specificity of OAS1 and suggests that early control of SARS-CoV-2 replication through OAS1 p46 is an important determinant of COVID-19 severity.

## Introduction

Oligoadenylate synthetase (OAS) proteins are a family of interferon (IFN)-induced sensors of viral RNA critical for cell-intrinsic innate immune defense against viruses through activation of the latent endoribonuclease RNase L (*Hornung et al., 2014*). Recognition of viral double-stranded RNA (dsRNA) induces a conformational change in OAS proteins to reveal a catalytic pocket which converts ATP to the second messenger $2'-5'A$ (*Lohöfener et al., 2015*). Binding to $2'-5'A$ dimerizes and activates RNase L, which cleaves cellular and viral RNA in order to block viral replication (*Han et al., 2014*). Although the importance of RNase L in restricting a variety of viruses is well documented, it is unclear how individual OAS proteins contribute to this breadth of antiviral activity (*Silverman, 2007*). Interestingly, the C-terminal region of human *OAS1* is alternatively spliced to produce the protein isoforms p42, p44, p46, p48, and p52, named according to their molecular weight (*Bonnevie-Nielsen et al., 2005*). All human OAS1 isoforms share the first five exons of *OAS1*, which contain the RNA binding and catalytic domains, but each isoform splices a distinct sixth exon to generate unique C-termini. The specific antiviral roles of individual human OAS1 isoforms are not defined. The production of OAS1 p46 is controlled by an SNP in the splice-acceptor site of exon six in *OAS1* (rs10774671 A>G) (*Figure 1A*). This splice site SNP in OAS1 is associated with genetic susceptibility to multiple flaviviruses and autoimmune disorders (*El Awady et al., 2011; Haralambieva et al., 2011; Liu et al., 2017; Simon-Loriere et al., 2015*). The G allele of this SNP shifts the splice acceptor site in exon 6 by one nucleotide to generate p46, while other OAS1 isoforms, primarily p42, are produced when the A allele is present (*Lim et al., 2009*). OAS1 p46 is unique among OAS1 isoforms because it is the only isoform with a C-terminal CaaX (cysteine-aliphatic-aliphatic-any residue) motif. Proteins containing CaaX motifs at their C-termini undergo a post translational lipidation modification termed prenylation and are targeted to the cytosolic face of intracellular organelle membranes of the endomembrane system following post-prenylation processing at the endoplasmic reticulum (*Wang and Casey, 2016*). The significance of the CaaX motif in OAS1 p46 and whether endomembrane targeting might alter the antiviral activity of OAS1 is unknown.

How the subcellular targeting of OAS family members impacts their specificity and antiviral activity is unclear. Most intracellular viral RNA (vRNA) sensors localize to the cytosol where they are poised to sense accumulating vRNA during infection (*Ablasser and Hur, 2020; Chan and Gack, 2016*). However, since many RNA viruses replicate their RNA in close association with intracellular membranes, placing OAS proteins at membranous compartments may augment vRNA sensing in certain contexts. Notably, positive-strand RNA viruses, such as flaviviruses, picornaviruses, and coronaviruses replicate their RNA within modified host organelles of the endomembrane system (*Romero-Brey and Bartenschlager, 2014*). These replication organelles shield vRNA from detection by cytosolic RNA sensors such as RIG-I (*Neufeldt et al., 2016*). Whether or not the host has evolved strategies to survey specific intracellular membranes for viral replication is unclear. We therefore hypothesized OAS1 p46, through its CaaX motif, is targeted to the endomembrane system and this targeting gives it enhanced access to vRNA during infection. In support of this model, we show the prenylated isoform, OAS1 p46, is targeted to Golgi membranes and this membrane-targeting results in enhanced detection of vRNA and augmented antiviral activity against positive-strand RNA viruses such as flaviviruses, picornaviruses, and SARS-CoV-2. Our human genetic data further supports the contribution of *OAS1* rs10774671 to severity of COVID-19. More broadly, this work reveals how intracellular membrane targeting of OAS1 is critical for detecting human pathogenic viruses that replicate on organellar membranes.

## Materials and methods

### Cells, cell culture conditions, and treatments

All cells (*Supplementary file 1*) were incubated at 37°C with 5% $CO_2$. HEK293T, A549, Vero, PH5CH8, Huh7, HeLa, and primary human fibroblasts were grown in DMEM (Sigma) containing 10% heat-inactivated fetal bovine serum (FBS) (Atlanta Biologicals) and 1% penicillin-streptomycin-

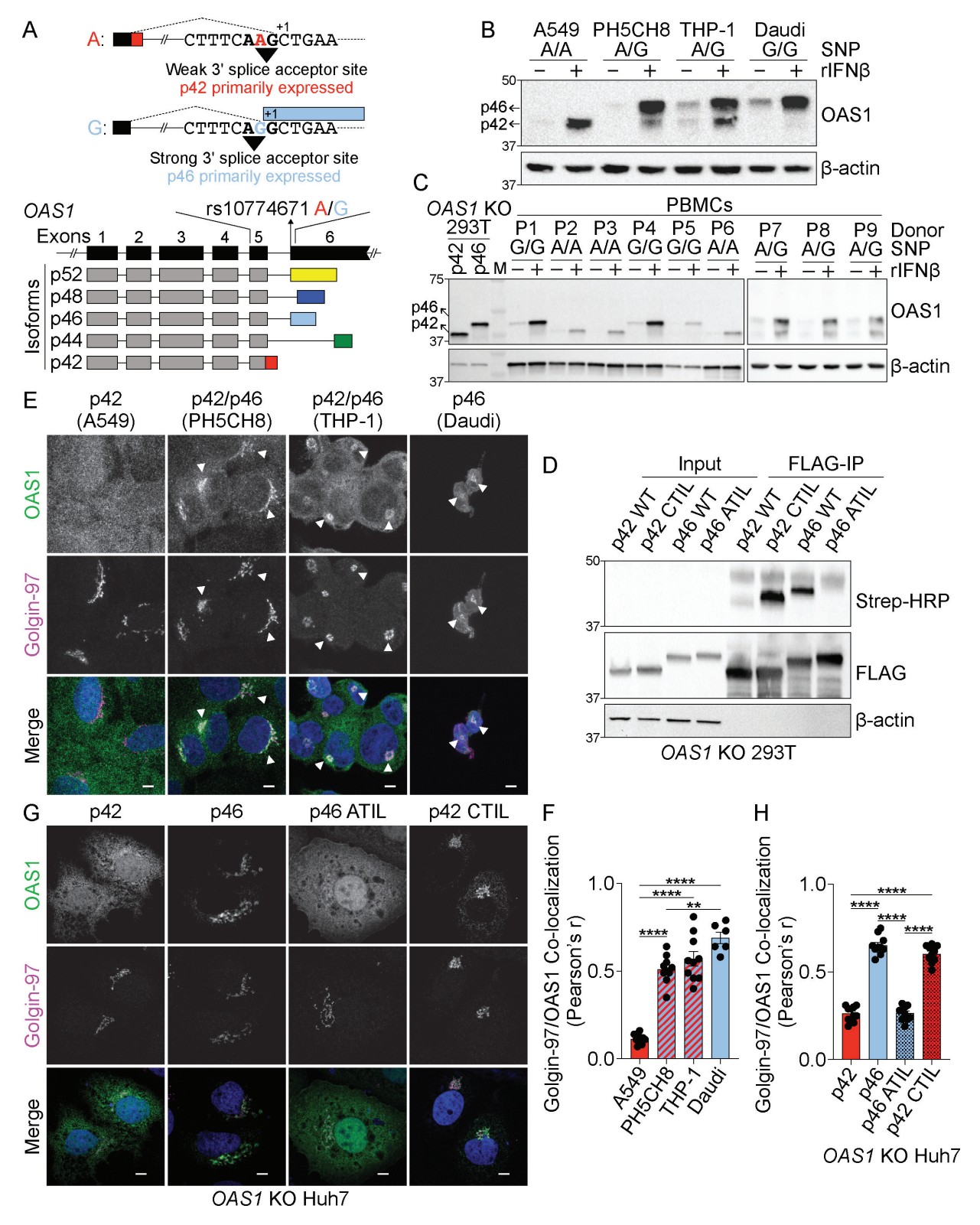

**Figure 1.** The p46 isoform of OAS1 is targeted to the endomembrane system. (**A**) Differential C-terminal splicing of OAS1 creates isoform diversity. (**B**) Immunoblot analysis of OAS1 isoform expression across cell lines treated with 1000 U/mL rIFNβ for 24 hr (n=3). (**C**) Immunoblot analysis of OAS1 isoform expression in PBMCs from donors with indicated genotype at rs10774671 treated with 1000 U/mL rIFNβ for 24 hr. Ectopic expression of OAS1 p42 and p46 in *OAS1* KO 293 T cells serves as control (three independent donors of each genotype depicted). (**D**) Immunoblot of *OAS1* KO 293T whole

*Figure 1 continued on next page*

*Figure 1 continued*

cell lysate (left) and immunoprecipitated (right) FLAG-tagged p42, p46, p42CTIL, or p46ATIL constructs subjected to Click-chemistry reaction with geranylgeranyl azide and alkyne biotin; representative immunoblot of two independent experiments is shown. (E) Representative maximum intensity projections of the indicated cell lines treated with 1000 U/mL with rIFNβ for 24 hr followed by staining with anti-OAS1 antibody (green), anti-Golgin-97 (magenta), and DAPI (blue); representative cells from one out of three independently performed experiments are depicted. (F) Pearson's correlation of OAS1 and Golgin-97 in individual cells from the indicated cell lines; each data point represents an individual cell from one representative experiment. (G) Representative confocal micrographs of *OAS1* KO Huh7 transfected with constructs encoding p42, p46, p42CTIL, or p46ATIL stained with anti-OAS1 (green) and Golgin-97 (magenta) antibodies and DAPI (blue); (n=2). (H) Pearson's correlation of OAS1 and Golgin-97 in *OAS1* KO Huh7 cells expressing p42, p46, p42CTIL, or p46ATIL; each data point represents an individual cell of one representative experiment. Scale on micrographs in (E) and (G) = 5 µm. (F, H) Data were analyzed using one-way ANOVA with Tukey's multiple comparisons test where **p<0.01, ****p<0001.

The online version of this article includes the following source data and figure supplement(s) for figure 1:

**Source data 1.** Uncropped gels for the associated panels in *Figure 1* and *Figure 1—figure suppleent 2*.
**Figure supplement 1.** Alignment and sequence analysis of OAS1.
**Figure supplement 2.** Biochemical and confocal analysis of human OAS1 isoforms.

glutamine (Mediatech). Daudi cells and PBMCs were cultured in RPMI 1640 (Sigma) containing 10% heat-inactivated FBS (Atlanta Biologicals) and 1% penicillin-streptomycin-glutamine (Mediatech). THP-1 cells were cultured in RPMI 1640 (Sigma) containing 10% heat-inactivated FBS (Atlanta Biologicals), 1% penicillin-streptomycin-glutamine (Mediatech), 10 mM Hepes (Corning), 1x NEAA (Corning), 1 mM sodium pyruvate (Corning), and 50 µM 2-mercaptoethanol (Sigma). Where applicable, THP-1 cells were differentiated in THP-1 media containing 40 nM of PMA (Sigma-Aldrich) for 24 hr. Recombinant human rIFNβ (PBL Interferon Source) was used at 200–1000 IU/mL. Primary human fibroblast lines were isolated from minced skin punch biopsies and subcultured to separate from epithelial and other cell types while grown in RPMI 1640 with 10% fetal calf serum supplemented with penicillin-streptomycin and gentamicin. Subculturing was performed with serial trypsinization and passage. Samples were consented under Seattle Children's Institutional Review Board (protocol #11738). Cryopreserved PBMCs from healthy control subjects genotyped for OAS1 rs10774671 were obtained from the Benaroya Research Institute Immune Mediated Registry and Repository. All subjects provided informed consent and the study was approved by the Institutional Review Board (IRB07109). PBMC were isolated from heparinized peripheral blood by Ficoll-Hypaque centrifugation. All cell lines used were tested negative for mycoplasma and none of the cell lines belong to the group of commonly misidentified cell lines according to the ICLAC register. Cell lines were authenticated by microscopy and through ligand/PAMP specificity.

## Generation of knockout cell lines using CRISPR/Cas9 gene editing

Cloning of OAS1 targeting guide RNA (gRNA) 5′-GTGCATGCGGAAACACGTGTCTGG-3′ into pRRLU6-empty-gRNA-MND-Cas7-t2A-Puro vector or RNase L targeting gRNA 5′-GTTATCC TCGCAGCGATTGCGGGG-3′ into pRRLU6-empty-gRNA-MND-Cas9-t2A-Blast was achieved using the In-Fusion enzyme mix (Clontech). *OAS1* and *RNASEL* KO 293T were generated using lentiviral transduction as described previously followed by selection in 2 µg/mL puromycin or blasticidin (*Lau et al., 2015*). Transient transfection was utilized to knockout *OAS1* in A549 and Huh7 cells. Cells were transfected with *OAS1* gRNA or a Cas9-expressing control vector using *Trans*IT-X2 (Mirus Bio) according to the manufacturer's instructions. At 24 hr post transfection, cells were selected with 2 µg/mL puromycin. Knockouts were validated by western blotting.

## Generation of 293T-ACE2 cells

Lentiviral expression vector for ACE2 (pLEX-ACE2) was generated by amplifying the *ACE2* sequence from cDNA from Huh7 cells (5′- GACTCTACTAGAGGATCCGCCACCATGTCAAGCTCTTCCTGGC TCC-3′ and 5′- GGGCCCTCTAGACTCGAGCTAAAAGGAGGTCTGAACATCATCAGTG-3′). This amplicon was cloned into a pLEX lentiviral backbone cut with *BamHI* and *XhoI* using the InFusion HD kit (Takara). pLEX-ACE2 was co-transfected with psPAX2 and pMD2.G into 293FT cells for lentiviral packaging. 293 T cells were transduced with ACE2-expressing lentivirus and selected with puromycin (2 mg/mL) for 4 days to generate 293T-ACE2 cells. ACE2 expression was verified by immunoblot (Proteintech).

## Immunoblotting

Cells were lysed in RIPA buffer (+1× HALT protease and phosphatase inhibitor), and 10–30 µg total protein from whole-cell lysates was run on SDS–PAGE and transferred to polyvinylidene difluoride membranes (Thermo Scientific). The membranes were blocked in 5% milk in PBS-T (pPBS/Tween 20). Primary antibody (*Supplementary file 2*) incubation with antibodies against OAS1 (CST), RNase L (CST), or FLAG (Sigma) were performed in 5% milk in PBS-T overnight at 4℃. Membranes were washed for 5 min in PBS-T three times. Secondary antibody incubation was performed in 5% milk in PBS-T at room temperature for 1 hr and after membranes were washed for 5 min in PBS-T three times. Membranes were imaged on a Chemidoc XR system.

## OAS1 siRNA knockdown

Dicer-substrate short interfering RNAs against a common region of *OAS1* or the unique 3'UTR of p42 or p46 were custom designed and procured from Integrated DNA Technologies (*Supplementary file 3*). THP-1 cells were differentiated with PMA for 24 hr. At 24 hr post treatment, cells were transfected with 20 nM of siRNA using *Trans*IT-X2 (Mirus Bio) according to the manufacturer's instructions. Viral infections were performed at 24 hr post-transfection.

## Cloning

Expression plasmids encoding OAS1 p42, p42CTIL, p44, p46, p46ATIL, p48, and p52 were generated by Gibson assembly of the common OAS1 sequence with the isoform-specific sequence into the pCDNA3.1 vector (*Supplementary file 3*). Empty pcDNA3.1 was cut using *BamHI* and *XbaI*. A Gibson assembly compatible fragment for the common sequence of OAS1 was PCR-amplified from an OAS1 expression plasmid (gift from Dan Stetson) using primers 5'-TGGTACCGAGCTCGATGATGGATCTCAGAA-3' and 5'-CAGCAGAATCCAGG AGCTCACTGG-3'. Gibson assembly compatible fragments for the unique sequences of p42, p42CTIL, and p44 were generated by PCR amplification of annealed sense and antisense oligos (*Supplementary file 3*). Gibson assembly compatible sequences for the unique portions for p46, p46ATIL, and p48, were generated by PCR amplification of gBlocks. N-terminal FLAG-tagged versions of OAS1 p42 and p46 were generated by cutting pcDNA3.1 OAS1 p42, p42CTIL, p46, and p46ATIL with *BamHI* and cloning of a 3xFLAG fragment by PCR amplification from pEF FLAG-ZAP-L (*Schwerk et al., 2019*) and Gibson assembly using primers 5'-CGACTCACTATAGGGAGACCCAAGCTTGGTACCGAGCTCGATGGACTACAAAGAC-3' and 5'-GTCCAGAGATTTGGCTGGGGTATTTCTG AGATCCATCATGCTTGTCATCGTCATCCTTGTAATCGATG-3' (*Supplementary file 3*). (*Schwerk et al., 2019*) and Gibson assembly using primers 5'-CGACTCACTATAGGGAGACCCA AGCTTGGTACCGAGCTCGATGGACTACAAAGAC-3' and 5'–GTCCAGAGATTTGGCTGGGGTATTTCTGAGATCCATCATGCTTGTCATCGT CATCCTTGTAATCGATG-3' (*Supplementary file 3*).

Expression plasmids encoding p42DADA, p46DADA in pCDNA3.1 were generated by site-directed mutagenesis on pCDNA3.1 p42 or p46. FLAG-tagged expression plasmids encoding p46 common+CTIL, p46 alanine mutant 9, p46 D12aa, p46 D22aa, p46 D32aa, p42 CFK mutant, and p46 CFK mutant were generated by site-directed mutagenesis on pCDNA3.1 FLAG-p46. Site-directed mutagenesis was performed using the QuikChange Lightning kit (Agilent) according to the manufacturer's instructions. All primers used for site-directed mutagenesis are listed in *Supplementary file 3*. OAS1 p46 C-terminal alanine mutants 1–8 and OAS1 p46 C-terminus species hybrids were generated by ligation of mutant gBlocks (*Supplementary file 3*). Briefly, pcDNA3.1 FLAG-OAS1 p46 was cut with *KflI* and *ApaI*. gBlocks were PCR-amplified using the following primer pair 5'-TAAGAA TTGGGATGGGTCCCCAG-3' and 5'-GACACTATAGAATAGGGCCCTCTAGA-3' and then cut with with *KflI* and *ApaI*. Ligation was performed using T4 DNA ligase (Thermo Fisher Scientific) according to the manufacturer's instructions.

## Geranylgeranyl click chemistry immunoprecipitation

Geranylgeranyl click chemistry IP reactions were performed using the Click-iT labeling kit and reagents (Thermo Fisher) according to the manufacturer's instructions with the following modifications. 293T *OAS1* KO cells were incubated with 25 µM geranylgeranylalcohol azide (GGAA) and transfected with 250 ng/mL pCDNA3.1 FLAG-tagged OAS1 expression constructs 3 hr after addition of GGAA. 24 hr after transfection, cells were lysed in Co-IP (50 mM Tris-HCl, pH 7.5; 150 mM NaCl;

0.5% NP40; 1 mM EDTA) buffer and OAS1 proteins were immunoprecipitated from the lysate using 20 µg anti-FLAG antibody (Sigma) and Dynabeads Protein G. After five washes in Co-IP buffer, the Dynabeads were resuspended in 50 µl 50 mM Tris-HCl, pH 8, and the click chemistry reaction was performed according to the manufacturer's instruction. Immunoprecipitated protein were immunoblotted and probed for presence of geranylgeranyl azide-biotin labeling using an HRP-conjugated streptavidin antibody.

## Virus infections and titer quantification

Virus and their sources are listed in *Supplementary file 4*. Encephalomyocarditis virus was grown and titered in Vero cells. West Nile virus Texas from Gale laboratory was grown as previously described (*Aarreberg et al., 2019*). CVB3-Nancy was prepared as previously described (*Laufman et al., 2019*). Influenza virus A/PR/8/34 and Influenza A virus A/Udorn/72 H3N2 R38A was prepared as described previously (*Min and Krug, 2006*). For EMCV, WNV, CVB, and IAV infections, *OAS1* KO 293 T cells were seeded in 12 well plates coated with 10 µg/mL poly-L-ornithine hydrobromide (Sigma) and allowed to adhere overnight. Plasmids were then transfected using *Trans*IT-X2 (Mirus Bio). At 24 hr post transfection, cells were infected with EMCV, WNV, or CVB at the indicated MOIs for 1 hr with gentle rocking. After 1 hr the inoculum was removed, and fresh media was added. For virus titer quantification, culture supernatants were serially diluted in DMEM using 96 well plates. For EMCV and WNV, titration was performed on Vero cells grown to 90% confluency in six-well plates. For CVB, titration was performed on HeLa cells grown to 90% confluency in six-well plates. IAV was titered on MDCK cells grown to 90% confluency in 12-well plates. The inoculum was removed after 1 hr of gentle rocking and a 0.8% agarose overlay was added containing the following: 0.8% UltraPure Low Melting Point Agarose (Thermo Fisher), 1x DMEM, 0.15% Sodium Bicarbonate, 10% heat-inactivated fetal bovine serum (FBS) (Atlanta Biologicals) and 1% penicillin-streptomycin-glutamine (Mediatech). For EMCV plates were fixed with 4% paraformaldehyde (Santa Cruz Biotechnology) at 24 hr post-infection followed by staining with a 5% crystal violet solution prepared by dissolving crystal violet (Sigma Aldrich) in a 50/50 mixture of 100% ethanol and deionized water. For IAV plates were fixed with 4% paraformaldehyde (Santa Cruz Biotechnology) at 24 hr post-infection followed by staining with a 5% crystal violet solution. For CVB plates were fixed with 4% paraformaldehyde (Santa Cruz Biotechnology) at 24 hr post-infection followed by staining with a 5% crystal violet solution. For WNV a neutral red overlay containing 0.01% neutral red (Sigma-Aldrich) 0.8% UltraPure Low Melting Point Agarose (Thermo Fisher), 1x DMEM, 0.15% sodium bicarbonate, 10% heat-inactivated fetal bovine serum (FBS) (Atlanta Biologicals) and 1% penicillin-streptomycin-glutamine (Mediatech) was added at 48 hr post-infection and plaques were read 24 hr later.

For quantification of Indiana vesiculovirus (VSV-GFP) replication, *OAS1* KO 293 T cells transfected with OAS1 p42, p46 and EV for 24 hr prior to infection, were infected with Indiana vesiculovirus (VSV-GFP; MOI=0.5, 6 hr). Cells were harvested using trypsin, washed 2x with PBS, and incubated with Zombie NIR fixable viability dye (1:1000) (BioLegend) for 30 min at room temperature in PBS. Cells were washed 2x with PBS, fixed in 4% PFA for 10 min at room temperature, and then washed once with FACS buffer (PBS with 0.1%BSA). Flow cytometry was performed on a BD FACS Canto II. % VSV-GFP+ cells were quantified using FlowJo (Tree Star).

SARS-CoV-2 strain USA/WA-1/2020 was propagated and titered on VeroE6 cells (gift of Dr. Ralph Baric). For infections, 293T-ACE2 cells seeded in 24-well plates (100,000 cells/well) were transfected with 250 ng of plasmid (Empty vector, p42, p46, and p46 ATIL) using the TransIT X2 kit (Mirus). A duplicate plate was transfected at the same time in order to confirm OAS1 expression by immunoblot. 24 hr post-transfection, cells were infected with SARS-CoV-2 at an MOI of 0.1 in serum-free DMEM for 1 hr, and media was replenished with DMEM containing 4% serum. Supernatants were harvested at 48hpi, and serial dilutions were tittered on Vero E6 cells seeded at 90% confluency in 12-well plates. Inoculum was removed after 1 hr of gentle rocking and replenished with an agarose overlay containing 0.4% Noble Agar (Thermo Fisher) in DMEM containing 10% serum. At 72 hpi, plates were fixed with 10% formaldehyde and stained with crystal violet solution (0.1% crystal violet and 20% methanol in water).

## Confocal laser scanning microscopy

For all microscopy experiments, cells were seeded on #1.5 12 mm coverslips (Bioscience Tools) coated with 10 µg/mL poly-L-ornithine hydrobromide (Sigma) and allowed to adhere overnight. All antibodies used in this study are listed in *Supplementary file 2*. For experiments testing endogenous OAS1 localization, cells were treated with rIFNβ for 24 hr. For experiments testing OAS1 localization in dox-inducible OAS1 A549 cells, cells were treated with 200 ng/mL doxycycline for 24 hr. For experiments testing localization of OAS1 in Huh7 cells, plasmids were transfected for 24 hr using Lipofectamine 3000 (Thermo Fisher) according to the manufacturer's instructions. At 24 hr post treatment/expression, cells were washed with PBS and then fixed in 4% PFA (Electron Microscopy Sciences) in PBS for 10 min at room temperature, washed with PBS, and then permeabilized with PBS containing 0.1% Triton X-100 for 10 min at room temperature. Cells were washed with PBS and then resuspended in a 3% BSA/PBS blocking solution for 1 hr. After blocking, cells were stained with rabbit anti-OAS1 (CST) and mouse IgG1 anti-Golgin 97 in PBS containing 1% BSA and 0.3% Triton X-100 for 1 hr in the dark at room temperature. Cells were washed three times with PBS and then stained with the secondary antibodies goat anti-rabbit IgG Alexa Fluor 488 (Themo Fisher) and goat anti-mouse IgG Alexa Fluor 648 (Thermo Fisher) in PBS containing 1% BSA and 0.3% Triton X-100 for 1 hr in the dark at room temperature. Samples were washed once with PBS, stained with DAPI in PBS for 10 min in the dark, followed by washing three times with PBS and then mounted with Pro-Long Glass antifade mounting media (Thermo Fisher). Samples were cured in the dark at room temperature for 24–48 hr prior to imaging.

For experiments testing OAS1 localization during viral infections, OAS1 was expressed as described above. At 24 hr post transfection, cells were infected with the indicated virus for 1 hr with gentle rocking followed by removal of the inoculum and replacement with fresh media. At 24 hr post treatment/expression, cells were washed with PBS and then fixed in 4% PFA (Electron Microscopy Sciences) in PBS for 10 min at room temperature, washed with PBS, and then permeabilized with PBS containing 0.1% Triton X-100 for 10 min at room temperature. Cells were washed with PBS and then resuspended in a 3% BSA/PBS blocking solution for 1 hr. After blocking, cells infected with EMCV were stained with rabbit anti-OAS1 (CST) and mouse IgG1 anti-dsRNA 9D5 in PBS containing 1% BSA and 0.3% Triton X-100 for 1 hr in the dark at room temperature. Cells infected with WNV were stained with rabbit anti-OAS1 (CST), mouse IgG1 anti Golgin 97 (CST) or mouse anti IgG1 PDIA3 (Sigma-Aldrich) and mouse IgG2a anti-dsRNA J2 (Scicons) in PBS containing 1% BSA and 0.3% Triton X-100 for 1 hr in the dark at room temperature. EMCV infected cells were washed three times with PBS and then stained with the secondary antibodies goat anti-rabbit IgG Alexa Fluor 488 (Themo Fisher) and goat anti-mouse IgG Alexa Fluor 648 (Thermo Fisher) in PBS containing 1% BSA and 0.3% Triton X-100 for 1 hr in the dark at room temperature. WNV infected cells were washed three times with PBS and then stained with the secondary antibodies goat anti-rabbit IgG Alexa Fluor 488 (Themo Fisher), goat anti-mouse IgG1 (Thermo Fisher), and goat anti-mouse IgG2a Alexa Fluor 648 (Thermo Fisher) in PBS containing 1% BSA and 0.3% Triton X-100 for 1 hr in the dark at room temperature. Samples were washed once with PBS, stained with DAPI in PBS for 10 min in the dark, followed by washing three times with PBS and then mounted with ProLong Glass antifade mounting media (Thermo Fisher). Samples were cured in the dark at room temperature for 24–48 hr prior to imaging. Samples were imaged using a Nikon Eclipse Ti laser scanning confocal microscope using a 60x oil-immersion lens. Images were processed and analyzed using the NIS elements software and Fiji. Quantification of co-localization was performed using the Fiji Coloc two plugin.

## RNA isolation, reverse transcription, and RT-qPCR

Total RNA was isolated using the NucleoSpin RNA kit (Macherey-Nagel) according to the manufacturer's protocol. cDNA was synthesized from 1 µg total RNA using the QuantiTect RT kit (Qiagen) according to the manufacturer's instructions. RT-qPCR was carried out using the ViiA7 RT-qPCR system with *Taq*Man reagents using *Taq*Man primers/probes (Life Technologies) for EMCV 5′UTR (*Supplementary file 3*).

## RNA immunoprecipitation

*OAS1* KO Huh7 cells were seeded the day before transfection with FLAG-p42, FLAG-p42CTIL, FLAG-p46, FLAG-p46ATIL or an EV control using *Trans*IT-X2 (Mirus Bio). At 24 hr post transfection,

cells were infected with EMCV at an MOI of 0.001. At 12 hr post-infection, cells were harvested and lysed in RNA-IP lysis buffer (100 mM KCl, 5 mM $MgCl_2$, 10 mM HEPES pH 7.4, 0.5% NP-40, 1 mM DTT, 1× HALT protease inhibitor, 100 U/mL RNasin and 2 mM ribonucleoside-vanadyl complex). Nuclei and debris were removed from the cytosolic lysate by centrifugation at 8000 $g$ at 4°C for 10 min. Next, 400 µg protein from the cytosolic lysate was incubated with 5 µg anti-FLAG mouse IgG1 (M2, Sigma) or mouse IgG1 control overnight at 4°C, with rotation. The next day, 0.75 mg Dynabeads Protein G (Invitrogen) was added, and the lysate was incubated for 2 hr at 4°C with rotation. After washing, coprecipitated RNA was isolated from IgG1-protein complexes by chloroform-iso-amyl alcohol extraction, reverse transcribed into cDNA (QuantiTect RT kit, Qiagen) and analyzed by RT-qPCR.

## OAS1 in vitro activity assay

Enrichment of FLAG-tagged OAS1 isoform proteins prior to in vitro activity assay was performed as described for the RNA-IP above. Briefly, *OAS1* KO 293 T cells were seeded on a 10 cm dish and transfected with FLAG-tagged OAS1 isoform expression plasmids and harvested 24 hr post transfection in Co-IP lysis buffer (50 mM Tris-HCl pH 8, 150 mM NaCl, 0.5% Igepal Ca-630, 1 mM EDTA, 1x HALT protease inhibitor). Lysate was incubated with 10 µg anti-FLAG mouse IgG1 (M2, Sigma) at 4°C, with rotation. The next day, 1.5 mg Dynabeads Protein G (Invitrogen) was added, and the lysate was incubated for 2 hr at 4°C with rotation, and then washed 6x with Co-IP lysis buffer. Immunoprecipitated protein samples were incubated in reaction buffer (10 mM Tris-HCl pH 7.5, 25 mM NaCl, 10 mM $MgCl_2$, 1 mM DTT, 0.1 mg/mL BSA) supplemented with 400 mM ATP, ~80 nM [a-$^{32}$P]-labeled ATP (3000 Ci/mmol 10 mCi/mL, 250 µCi; PerkinElmer), and 33.3 µg/mL poly(I:C) (Invivogen). The reactions were allowed to proceed for 2 hr at 37°C. The reactions were then analyzed by denaturing gel electrophoresis on a 20 cm tall 20% polyacrylamide 7 M urea gel with 0.5 x TBE running buffer at 12.5 W. The gels were then applied onto Whatman filter paper, covered with plastic, and exposed directly to a PhosphorImager screen (GE Healthcare) for 15 to 40 min, as necessary. [$^{32}$P]-labeled $2'{-}5'$ oligoadenylate products were visualized using a Sapphire Biomolecular Imager (Azure Biosystems).

## Genetic association

Samples from a cohort of 34 severe COVID-19 cases were collected starting in April 2020 at Virginia Mason Medical Center and Benaroya Research Institute. Severity was based on hospitalization in the critical care unit with mechanical ventilation. A cohort of 99 healthy control subjects matched for ancestry (self-reported) was assembled from participants in the healthy control registry at Benaroya Research Institute. Both studies were approved by the Institutional Review Board at Benaroya Research Institute (IRB20-036 and IRB07109 respectively). A description of the cohorts is presented in *Supplementary file 5*. DNA samples from these subjects were genotyped for *OAS1* rs10774671 using a Taqman SNP genotyping assay (Thermo Fisher). Genotypes passed Hardy-Weinberg equilibrium analysis. Association testing was performed using gPLINK v2.050 (https://zzz.bwh.harvard.edu/plink/gplink.shtmll) by logistic regression adjusting for sex and ancestry (race/ethnicity). Replication of genetic association was tested using 1676 critically ill COVID-19 cases collected through the GenOMICC study in the UK and 8380 population-based controls (1:5 cases:controls) from the UK Biobank samples as described (*Pairo-Castineira et al., 2021*). All subjects were of European descent as determined by ancestry informative markers. DNA was genotyped using the Illumina Global Screening Array v3.0+ multi disease bead chips (Illumina) and subjected to standard quality control filters. Severity of local COVID-19 patients was classified based on maximum disease severity using a 7-point ordinal scale described previously (*Cao et al., 2020*). 1, not hospitalized with resumption of normal activities; 2, not hospitalized, but unable to resume normal activities; 3, hospitalized, not requiring supplemental oxygen; 4, hospitalized, requiring supplemental oxygen; 5, hospitalized, requiring nasal high-flow oxygen therapy, noninvasive mechanical ventilation, or both; 6, hospitalized, invasive mechanical ventilation; and 7, death. COVID-19 patients with ordinal scores of 6 and 7 were included in the genetic analysis. LD of the two SNPs was assessed from 1000 Genomes http://uswest.ensembl.org/Homo_sapiens/Info/Index. The summary association data for the replication cohort is included in https://genomicc.org.data (*Pairo-Castineira et al., 2021*). Tests for association between cases and controls were performed by logistic regression using PLINK, with sex, age (as of

April 1, 2020), deprivation score of residential postal code, and the first 10 principal components as covariates.

## Statistics

Statistical analyses (other than genetic analysis) were performed with Prism eight and the specific statistical analyses performed are indicated in the figure legends.

## Results

### OAS1 p42 and p46 are the major OAS1 isoforms

Alignment of the C-terminal regions of the five OAS1 isoforms revealed a C-terminal CaaX motif present only in the p46 isoform, which may target this isoform to intracellular membranes (*Figure 1A*, *Figure 1—figure supplements 1A* and *2A*). OAS1 proteins from diverse vertebrate species have a CaaX motif at their C-termini, suggesting the CaaX motif has a conserved role in the function of OAS1 (*Figure 1—figure supplement 1C*). The G allele is not evenly distributed across human populations and is more prevalent in individuals of African descent (*Figure 1—figure supplement 1B*). We determined the protein expression of OAS1 isoforms across several human cell lines and assessed the impact of the SNP (rs10774671) on the expression of OAS1 isoforms. Genotyping revealed that the A549 cell line carried only the A allele, that PH5CH8 and THP-1 cell lines carried both the A and G alleles, and that Daudi cells carried only the G allele at rs10774671 (*Figure 1B*). We treated these cell lines with recombinant human interferon beta (rIFNβ) or Sendai virus and evaluated OAS1 protein expression by immunoblot using an N-terminal OAS1 antibody that recognizes all isoforms. We detected two OAS1 isoforms at molecular weights of 42 and 46 kDa in cells with at least one copy of the G allele, while only p42 or p46 were expressed in A/A or G/G cells, respectively (*Figure 1B* and *Figure 1—figure supplement 2B*). We did not observe the expression of other OAS1 isoforms under these conditions. We confirmed this SNP-dependent expression in human peripheral blood mononuclear cells (PBMCs) homozygous for either allele at rs10774671 after treatment with rIFNβ for 24 hr. Cells homozygous for the A allele only produced p42, while cells homozygous for the G allele only produced p46 (*Figure 1C*). *OAS1* KO 293 T cells generated by CRISPR/Cas9 gene editing (*Figure 1—figure supplement 2C*) were used to ectopically express OAS1 p42 and p46 as positive controls (*Figure 1C*). Corroborating our findings, PBMCs harboring at least one G allele expressed p46 stronger than p42, which may be influenced by splice site preference or mRNA stability. We did not observe expression of the p44, p48, or p52 isoforms in any of the cells used in this study, which is consistent with previous reports demonstrating that these isoforms are weakly expressed at the mRNA level, and unstable at the protein level (*Carey et al., 2019*; *Li et al., 2017*).

### The OAS1 p46 isoform is geranylgeranylated

One of the major differences between OAS1 p42 and p46 is the presence of a C-terminal CaaX motif in p46 (*Figure 1—figure supplements 1A* and *2A*). Proteins containing CaaX motifs at their C-termini are prenylated at the ER and are targeted to the cytosolic face of cellular membranes of the endomembrane system, including the endoplasmic reticulum and Golgi apparatus (*Wang and Casey, 2016*). Depending on the identity of the 'X' amino acid, CaaX-containing proteins can be either farnesylated or geranylgeranylated by the respective farnesyl or geranylgeranyl transferases. CaaX motifs with a leucine in the 'X' position, as in the case of p46, are preferentially geranylgeranylated by geranylgeranyl transferase I (GGTaseI) (*Hartman et al., 2005*). In order to test if the CaaX motif in p46 is geranylgeranylated, we performed an in vitro geranylgeranylation detection assay using a click chemistry approach (*DeGraw et al., 2010*). For this, *OAS1* KO 293 T cells were transfected with N-terminal FLAG-tagged OAS1 p42, p46, p42CTIL, and p46ATIL expression constructs. The p42CTIL construct represents p42 with the addition of the CaaX (CTIL) motif from p46, which allowed us to test if the CaaX motif from p46 is sufficient to drive geranylgeranylation of p42. The p46ATIL construct was generated by mutating the CaaX cysteine to an alanine (C>A) to disrupt its prenylation, allowing us to test if this motif is necessary for its geranylgeranylation. Cells were incubated with a geranylgeranyl azide alcohol substrate, which is incorporated into all geranylgeranylated proteins in the cell and serves as a tag in the subsequent click-chemistry reaction with a biotin

alkyne. FLAG-tagged OAS1 protein isoforms were immunoprecipitated from whole cell lysate and subjected to on-bead click-chemistry reaction, which forms a covalent bond between the azide and alkyne moieties and labels geranylgeranylated proteins with a biotin tag, which was then probed for. Immunoblotting of the immunoprecipitated OAS1 proteins revealed CaaX-containing p46 and p42CTIL proteins were geranylgeranylated, while p42 and p46ATIL were not (*Figure 1D*). These data confirm that p46, but not p42, is geranylgeranylated and demonstrate that the CaaX motif is both necessary and sufficient for the geranylgeranylation of OAS1.

## OAS1 p46 localizes to the endomembrane system

Based on the differential geranylgeranylation of the p42 and p46 OAS1 isoforms, we hypothesized that these proteins would localize to unique subcellular compartments. Since we verified which OAS1 isoforms are produced in PBMCs from specific donors and several human cell lines, we could stratify these cells into those producing p42 and or p46 based on the presence of an A or the G allele at rs10774671. PH5CH8, THP-1, and PBMCs containing both A and G alleles all produced p42 and p46, while PBMCs and A549 cells that carried only the A allele only produced p42 (*Figure 1B and C* and *Figure 2—figure supplement 1C*). We used these cells to evaluate if OAS1 isoforms localize to different compartments under endogenous expression conditions using confocal laser scanning microscopy (cLSM). We observed OAS1 localization at a perinuclear compartment in p46-producing cells after rIFNβ treatment, while, in contrast, OAS1 localized to the cytosol and nucleus in cells incapable of producing p46 (*Figure 1—figure supplement 2D* and *Figure 2—figure supplement 1C*). Co-staining with Golgin-97 identified that OAS1 was predominantly localized at the *trans*-Golgi network in p46-producing cells (*Figure 1E and F*, *Figure 1—figure supplement 2E* and *Figure 2—figure supplement 1C*). Cells producing p46 showed a significant increase in OAS1/Golgin-97 co-localization compared to A549 cells that only produce p42 (*Figure 1E and F*, *Figure 1—figure supplement 2E* and *Figure 2—figure supplement 1C*). Previous reports have suggested mitochondrial localization of OAS1 (*Kjær et al., 2014*; *Kjaer et al., 2009*). However, we were not able to confirm these findings. Our data revealed that OAS1 localizes to intracellular organellar membranes in p46 isoform expressing cells.

We confirmed the distinct localization of OAS1 p46 and p42 by performing cLSM on *OAS1* KO Huh7 cells ectopically expressing these isoforms (*Figure 1G and H* and *Figure 1—figure supplement 2F*). OAS1 localized to the Golgi in cells ectopically expressing p46, while the p42 isoform localized to the cytosol and the nucleus (*Figure 1G and H*). Compared with p42, p46 showed significantly stronger localization to the Golgi (*Figure 1G and H*). We next tested the contribution of the CaaX motif to the Golgi localization of p46 by expressing the CaaX mutant p46ATIL (C>A) and evaluating its localization by microscopy. p46ATIL localized to the cytosol and nucleus, similar to p42, indicating the CaaX motif is necessary for the Golgi localization of p46 (*Figure 1G and H*). We then tested if adding the CaaX motif to the p42 isoform is sufficient to localize p42CTIL to the Golgi. Consistent with our hypothesis, ectopically expressed p42CTIL localized to the Golgi and showed significantly stronger co-localization with the Golgi over p42, confirming the CaaX motif is both necessary and sufficient to localize OAS1 isoforms to the Golgi (*Figure 1G and H*). These findings were further confirmed in *OAS1* KO A549 cells ectopically expressing p42, p46, p42CTIL, and p46ATIL (*Figure 1—figure supplement 2G, H and I*). The p44, p48, and p52 isoforms, which also lack a CaaX motif, localized to the cytosol and nucleus in a manner similar to p42 when ectopically expressed in *OAS1* KO Huh7 cells (*Figure 1—figure supplement 2J and K*). These experiments demonstrate that the p46 isoform of OAS1 localizes to the endomembrane system, particularly the Golgi, in a prenylation-dependent manner.

## OAS1 p42 and p46 are differentially antiviral

The distinct localization of OAS1 isoforms led us to hypothesize that these isoforms have differential antiviral activity. Specifically, localization of p46 to the endomembrane system led us to hypothesize that this isoform may have enhanced antiviral activity against viruses that use these organelle membranes for replication. Encephalomyocarditis virus (EMCV) is a positive-strand RNA virus in the picornavirus family, which replicates on organelles of the endomembrane system, particularly ER and Golgi membranes, and is sensitive to the OAS/RNase L pathway (*Chebath et al., 1987*; *Melia et al., 2018*). We transfected plasmids encoding p42, p46, or an empty vector control in *OAS1* KO 293 T

cells and infected with EMCV at a multiplicity of infection (MOI) of 0.001 at 24 hr post-transfection (*Figure 2A*). At 24 hr post-infection, total RNA and culture supernatants were collected for RT-qPCR of EMCV RNA or plaque assay, respectively. We found p42 expression led to a significant fivefold reduction in EMCV RNA over control, while p46 expression led to a significant 50-fold reduction over EV and a significant 10-fold reduction over p42 at 24 hr post-infection (*Figure 2B*). Quantification of viral titer from supernatants by plaque assay showed p46 expression led to a significant 100-fold reduction in EMCV titer over control and a significant 50-fold reduction over p42. In contrast, the p42 isoform reduced EMCV titer by fivefold over control (*Figure 2C* and *Figure 2—figure supplement 1A*). We also compared the antiviral activity of p44, p48, and p52 against EMCV over a range of doses and found their antiviral activity was inferior to p46 (*Figure 2D and E*). These data demonstrate that, among OAS1 isoforms, OAS1 p46 confers the strongest antiviral activity against EMCV.

Next, we tested the ability of endogenously expressed OAS1 isoforms to restrict EMCV using an siRNA knockdown approach in THP-1 cells (A/G at rs10774671) which express both the p42 and p46 isoforms. We transfected PMA-differentiated THP-1 macrophages with siRNAs against both OAS1 isoforms, or p42 or p46 alone, and then infected with EMCV (*Figure 2F*). Depletion of total OAS1 or p46 led to a significant four-fold increase in viral titer compared to a non-targeting control (NC), whereas specific knockdown of p42 had no effect on viral titer compared to NC siRNA (*Figure 2G* and *Figure 2—figure supplement 1B*). Finally, we tested if the G allele at rs10774671 correlates with resistance to EMCV in primary human fibroblasts isolated from six donors (3 A/A, 3 A/G). Quantification of EMCV RNA and viral titer at 24 hr post-infection revealed that cells with at least one copy of the G allele had reduced levels of EMCV burden (*Figure 2H and I*). These data show major antiviral differences in OAS1 isoforms at endogenous levels of expression and support an important role for p46, but not p42, in restricting a virus that utilizes the endomembrane system for replication.

## The antiviral function of OAS1 isoforms requires catalytic activity and RNase L

The differences in the antiviral activity and localization of OAS1 p42 and p46 led us to hypothesize that these isoforms may utilize unique antiviral mechanisms. Both RNase L and $2'-5'A$ independent antiviral mechanisms have been documented for OAS proteins, but whether human OAS1 p42 and p46 isoforms differentially utilize $2'-5'A$ or RNase L is unknown (*Carey et al., 2019*; *Elbahesh et al., 2011*; *Kristiansen et al., 2010*; *Lin et al., 2009*). In order to ablate catalytic activity and test if both OAS1 isoforms required $2'-5'A$ synthetase activity to be antiviral, we mutated two key aspartic acid residues required for synthetase activity, D75 and D77, in the catalytic core of p42 and p46 to alanine (D75A; A76A; D77A) which we will hereafter refer to as catalytic mutant (*Sarkar et al., 1999*). We expressed these catalytically inactive OAS1 isoforms alongside their corresponding wild-type constructs in *OAS1* KO 293 T cells and tested their ability to restrict EMCV (*Figure 3A*). As before, expression of p42 and p46 reduced EMCV RNA and titer to different degrees. However, the catalytically inactive OAS1 constructs failed to restrict EMCV RNA or titer (*Figure 3B and C*). Furthermore, the catalytic mutants failed to reduce EMCV RNA compared to their wild-type counterparts over a range of protein expression levels (*Figure 3—figure supplement 1A and B*). These data indicate that the ability to synthesize $2'-5'A$ is essential for both OAS1 p42 and p46 to restrict EMCV.

We next tested if p42 and p46 require RNase L for their antiviral activity by expressing p42 or p46 in RNase L-deficient 293 T cells (*Figure 3D* and *Figure 3—figure supplement 1C*). Compared to non-targeted Cas9 control cells, expression of p42 and p46 in *RNASEL* KO 293 T cells had no impact on EMCV vRNA or titers (*Figure 3E and F*). Next, we tested if RNase L expression was sufficient to rescue the antiviral activity of OAS1 isoforms in these cells. *RNASEL* KO 293 T cells complemented with RNase L and expressing p46 showed a significant reduction in EMCV RNA and titer over control. Although not significant, complementing p42 expressing cells with RNase L showed a trend in reducing viral RNA and titer (*Figure 3—figure supplement 1D, E and F*). These experiments show that RNase L is required for the antiviral activity of OAS1 p42 and p46 against EMCV.

Although the RNA binding and catalytic domains are identical between the p42 and p46, we tested if the distinct C-terminal regions of these isoforms could affect their enzymatic activity. To this end, we performed $2'-5'A$ synthetase activity assays using p42, p46, and their respective CaaX

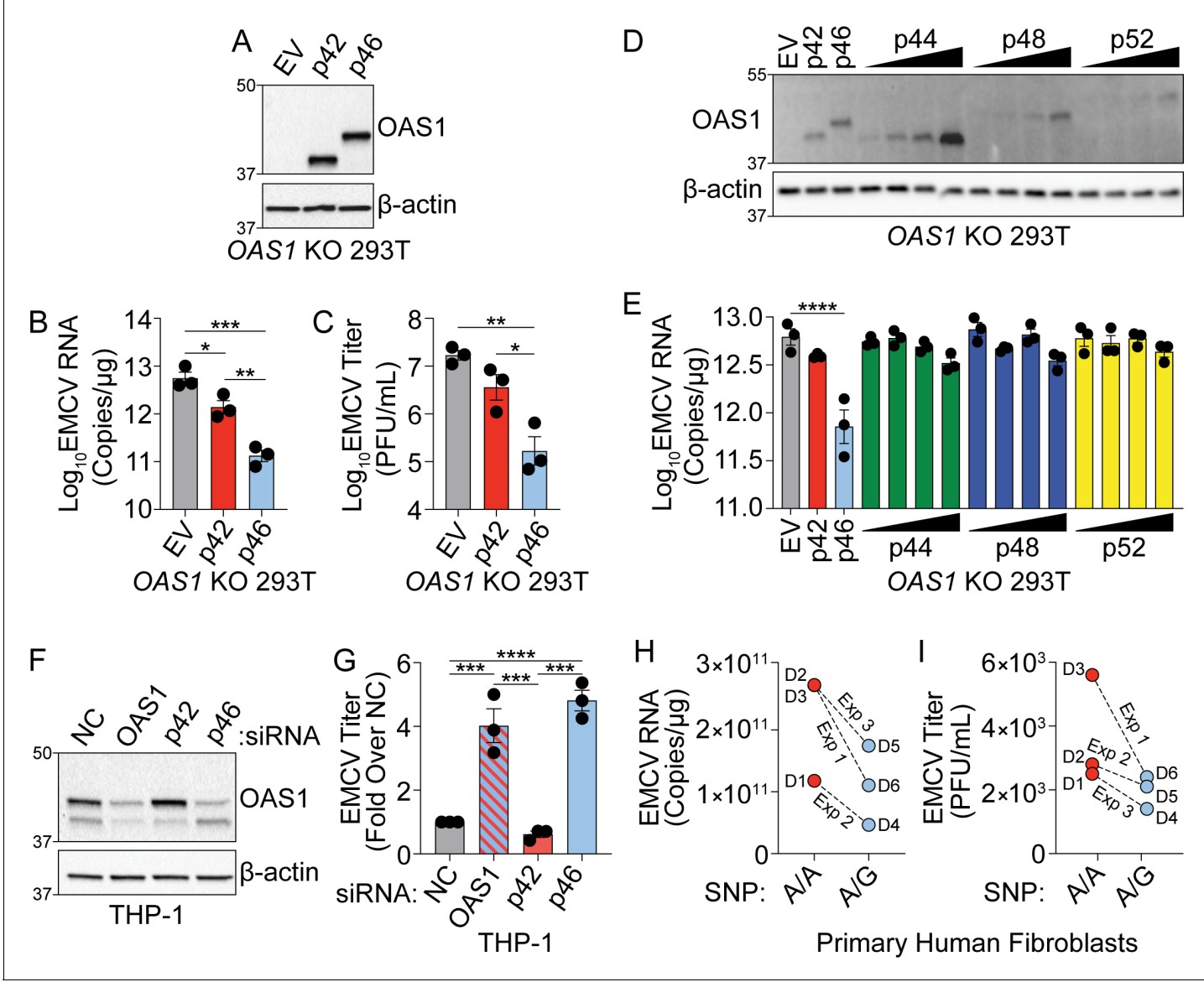

**Figure 2.** OAS1 isoforms are differentially antiviral. (**A**) Immunoblot analysis of p42 and p46 in *OAS1* KO 293 T cells at 24 hr post transfection. (**B**) Quantification of EMCV 5′UTR by RT-qPCR at 24 hr post-infection with EMCV (MOI=0.001) in *OAS1* KO 293 T cells transfected with a control EV, p42, and p46. (**C**) Viral titers at 24 hr post-infection with EMCV (MOI=0.001) in *OAS1* KO 293 T cells transfected with a control EV, p42, and p46. (**D**) Immunoblot analysis of OAS1 expression in *OAS1* KO 293 T cells at 24 hr post transfection with EV (500 ng), p42 (100 ng), p46 (100 ng), or 200, 350, or 500 ng of the corresponding catalytic mutant (500 ng DNA total in each transfection). (**E**) Quantification of EMCV 5′UTR in *OAS1* KO 293T transfected as in (**A**) for 24 hr followed by EMCV infection for 24 hr (MOI=0.001). (**F**) Immunoblot of OAS1 in PMA-differentiated THP-1 macrophages infected with EMCV (MOI=1, 24 hr) 24 hr post transfection with a non-targeting control siRNA (siNC) or siRNAs against total OAS1, p42, or p46. (**G**) Viral titers at 24 hr post EMCV infection (MOI=1) taken from PMA-differentiated THP-1 macrophages transfected with siNC, siOAS, sip42, or sip46. (**H**) Quantification of EMCV 5′UTR by RT-qPCR and (**I**) viral titers from primary human fibroblasts pre-treated with 25 U/mL rIFNβ for 24 hr prior to EMCV infection (MOI=0.01) for 24 hr; three independent experiments with paired donors of each genotype (A/A vs. A/G) are shown. (**B**, **C**), and (**G**) Data were analyzed using one-way ANOVA with Tukey's multiple comparisons test where *p<0.05, **p<0.01, ***p<0.001, ****p<0.0001. For (**E**) data were analyzed using a one-way ANOVA with Dunnett's multiple comparisons test (vs. EV) where ****p<0.0001. (**A**, **D**) and (**F**) Representative immunoblots of three independent experiments are shown. (**B**, **C**, **E**) and (**G**) Each data point represents an independently performed experiment. (**H**) and (**I**) Each data point represents an individual donor from three independently performed experiments (one donor pair per experiment).

The online version of this article includes the following source data and figure supplement(s) for figure 2:

**Source data 1.** Uncropped gels for the associated panels in *Figure 2*.

**Figure supplement 1.** Antiviral activity of human OAS1 isoforms.

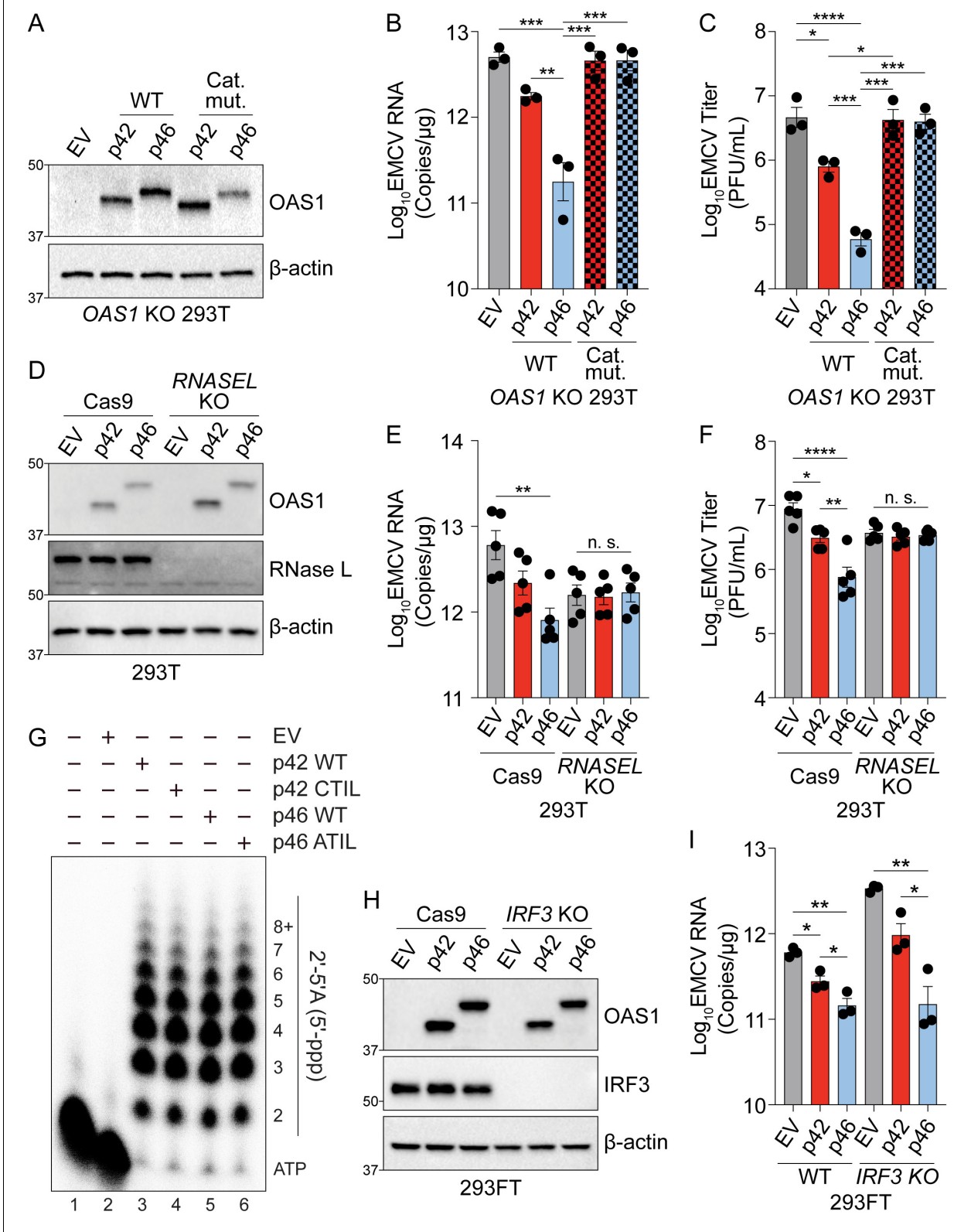

**Figure 3.** OAS1 isoforms require catalytic and RNase L activity. (**A**) Expression of OAS1 p42 and p46 along with their corresponding catalytic mutants (250 ng) at 24 hr post transfection in *OAS1* KO 293 T cells. (**B**) Quantification of EMCV 5′UTR by RT-qPCR in *OAS1* KO 293 T cells expressing a control EV, p42, p46, or their corresponding catalytic mutants (250 ng) at 24 hr post EMCV infection (MOI=0.001). (**C**) Viral titers at 24 hr post EMCV infection (MOI=0.001) taken from *OAS1* KO 293 T cells transfected with a control EV, p42, p46, or their corresponding catalytic mutants. (**D**) Immunoblot analysis

*Figure 3 continued on next page*

Figure 3 continued

of OAS1 and RNase L at 24 hr post transfection in Cas9 or *RNASEL* KO 293 T cells. (**E**) Quantification of EMCV 5′UTR by RT-qPCR in Cas9 and *RNASEL* KO 293 T cells expressing a control EV, p42, or p46 at 24 hr post EMCV infection (MOI=0.001). (**F**) Viral titers at 24 hr post-infection with EMCV (MOI=0.001) taken from Cas9 or *RNASEL* KO 293 T cells transfected with control EV, p42, or p46. (**G**) In vitro 2′−5′A synthesis assay of OAS1 p42 and p46 isoforms and their CaaX motif mutants; a representative blot of two independently performed experiments is depicted. (**H**) Immunoblot analysis of OAS1 and IRF3 in WT or *IRF3* KO 293FT cells at 24 hr post transfection. (**I**) Quantification of EMCV 5′UTR 24 hr post EMCV infection (MOI=0.001) in WT or *IRF3* KO 293FT cells transfected with a control EV, p42, or p46. (**B, C, E, F**) and (**I**) each data point represents an independently performed experiment. Data were analyzed using one-way ANOVA with Tukey's multiple comparisons test where *p<0.05, **p<0.01, ***p<0.001, ****p<0.0001. (**A, D**) and (**H**) Representative immunoblots of three (**A, H**) and five (**D**) independent experiments are shown.

The online version of this article includes the following source data and figure supplement(s) for figure 3:

**Source data 1.** Uncropped gels for the associated panels in *Figure 3* and *Figure 3—figure supplement 1*.
**Figure supplement 1.** RNase L-dependent activity of OAS1 isoforms.

mutants. As expected, these isoforms shared a similar capacity to synthesize 2′−5′A and the CaaX motif did not alter the catalytic activity (*Figure 3G*).

Cleavage of cellular RNA by RNase L has been shown to generate immunostimulatory RNAs that are sensed by RIG-I and MDA5 and increase antiviral protection through the production of interferon (*Malathi et al., 2007*). To determine if the differences in antiviral activity of p42 and p46 could be explained by differential production of immunostimulatory RNAs, we tested the antiviral activity of p42 and p46 in wild type or *IRF3* KO 293FT cells (*Figure 3H*). In wild type cells, overexpression of p42 and p46 reduced EMCV RNA significantly over the control. Importantly, loss of IRF3 expression did not impede the ability of p46 to restrict EMCV, as EMCV RNA was significantly reduced over control in *IRF3* KO cells expressing p46 (*Figure 3I*). Expression of p42 in *IRF3* KO cells showed a trend toward reducing EMCV RNA (*Figure 3I*). These data confirm p42 and p46 confer antiviral activity in the absence of additional factors that are induced by IRF3. Collectively, these data demonstrate that p42 and p46 both use the 2′−5′A/RNase L pathway and have a similar capacity to synthesize 2′−5′A to confer antiviral activity. This suggests that some other mechanism must account for their differential antiviral activities.

## Endomembrane targeting of p46 through the CaaX motif enhances access to viral RNA

Since the CaaX motif localizes p46 to the endomembrane system, we investigated if this motif played a role in the enhanced antiviral activity of p46 against EMCV. During picornavirus infection, ER- and Golgi-derived membranes form replication organelles that are sites of vRNA replication (*Melia et al., 2019*; *Melia et al., 2018*). Therefore, we hypothesized that endomembrane localization of p46 places this viral RNA sensor on membranes utilized by EMCV for replication and would allow the enhanced access of p46 to EMCV RNA. In contrast, p42 would have limited access to vRNA as it is localized to the cytosol and nucleus. We determined the localization of p42 and p46 during EMCV infection by infecting *OAS1* KO Huh7 cells expressing p42 or p46 with EMCV and then stained for OAS1 and dsRNA, followed by cLSM. In mock-infected cells expressing p46, this isoform was localized to perinuclear structures (*Figure 4A*). In cells infected with EMCV, p46 was no longer perinuclear and instead redistributed throughout the cell to sites of double-stranded viral RNA replication intermediate accumulation (*Figure 4A*). In contrast, p42 remained in the cytosol and nucleus before and after infection (*Figure 4A*). These data suggest that p46 is in close proximity to sites of EMCV RNA replication where dsRNA ligands are present. We next used an RNA immunoprecipitation approach to test whether OAS1 p46 has enhanced access to EMCV RNA and if the CaaX motif enhances its ability to bind EMCV RNA. Since RNase L cleaves RNA and is required for OAS1-mediated restriction of EMCV, RNase L-deficient Huh7 cells allowed us to test OAS1 isoform binding to EMCV RNA in samples with equivalent amounts of EMCV RNA (*Figure 4B*). N-terminally FLAG-tagged OAS1 p42, p46, p42CTIL, and p46ATIL isoforms were expressed in *OAS1* KO Huh7 cells, which are also naturally devoid of RNase L expression (*Figure 4—figure supplement 1A*). Expression of OAS1 isoforms in *OAS1* KO Huh7 cells had no impact on viral RNA levels (*Figure 4B*). However, RT-qPCR for EMCV RNA after FLAG immunoprecipitation revealed that p46 tended to bind more EMCV RNA than p42 (*Figure 4B*). Although adding a CaaX motif to p42 (p42CTIL) did not significantly increase the ability of p42 to pull down EMCV RNA, mutating the CaaX motif on p46

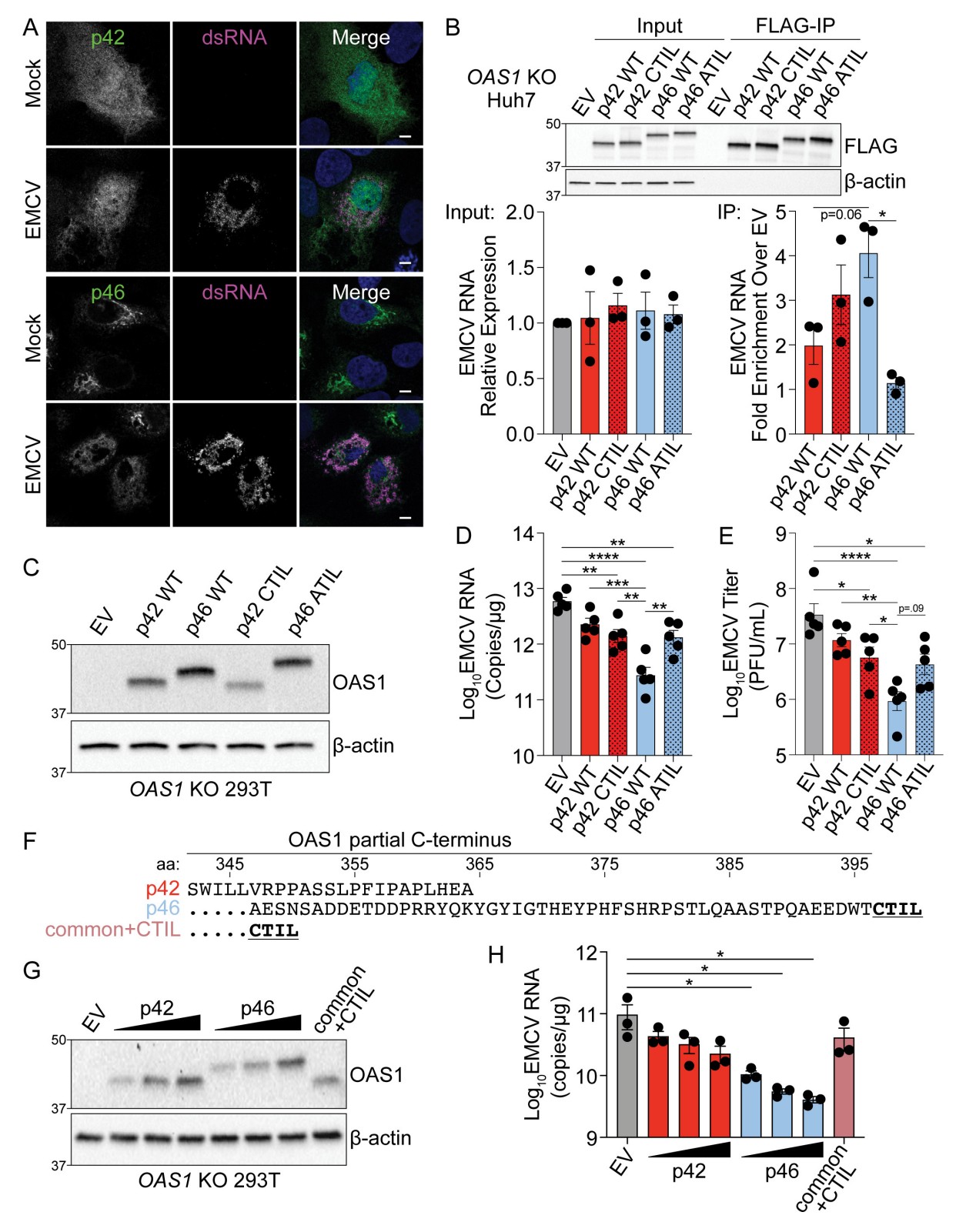

**Figure 4.** Endomembrane targeting of OAS1 p46 through the CaaX motif enhances access to viral RNA. (**A**) Confocal micrographs from mock or EMCV infected (MOI=0.001, 12 hr) *OAS1* KO Huh7 cells ectopically expressing p42 or p46 and stained with anti-OAS1 (green) and anti-dsRNA (magenta) antibodies and DAPI (blue); representative cells from one out of two independently performed experiments are depicted. (**B**) Immunoblot of FLAG immunoprecipitation performed on *OAS1* KO Huh7 cells expressing control EV, FLAG-p42, FLAG-p42CTIL, FLAG-p46, and FLAG-p46ATIL.

*Figure 4 continued on next page*

*Figure 4 continued*

Quantification of EMCV 5'UTR via RT-qPCR in the input or after RNA immunoprecipitation performed on *OAS1* KO Huh7 cells transfected with control EV, FLAG-p42, FLAG-p42CTIL, FLAG-p46, and FLAG-p46ATIL infected with EMCV (MOI=0.001, 12 hr). (C) Immunoblot analysis of p42, p46, p42CTIL, and p46ATIL in *OAS1* KO 293 T cells at 24 hr post transfection. (D) Quantification of EMCV 5'UTR by RT-qPCR from *OAS1* KO 293T expressing a control EV, p42, p46, p42CTIL, or p46ATIL at 24 hr post EMCV infection (MOI=0.001). (E) Viral titers quantified at 24 hr post-infection with EMCV at an MOI of 0.001 in *OAS1* KO 293 T cells transfected with control EV, p42, p46, p42CTIL, or p46ATIL. (F) Alignment of C-termini of expression constructs used in (G) and (H). (G) Immunoblot analysis of p42 (50 ng, 100 ng, 200 ng), p46 (50 ng, 100 ng, 200 ng) and common +CTIL (500 ng) in *OAS1* KO 293 T cells at 24 hr post transfection. (H) Quantification of EMCV 5'UTR by RT-qPCR from *OAS1* KO 293 T cells transfected as in (G) at 24 hr post EMCV infection (MOI=0.001). Scale = 5 μm. (B, D), and (E) Data were analyzed using one-way ANOVA with Tukey's multiple comparisons test where *$p<0.05$, **$p<0.01$, ***$p<0.001$, ****$p<000.1$. (H) Data were analyzed using a one-way ANOVA with Dunnett's multiple comparisons test (vs. EV) where *$p<0.05$. (B, C) and (G) Representative immunoblots of three (B, G) and five (C) independent experiments are shown. (B, D) and (H) Each data point represents an independently performed experiment.

The online version of this article includes the following source data and figure supplement(s) for figure 4:

**Source data 1.** Uncropped gels for the associated panels in *Figure 4* and *Figure 4—figure suppleent 1*.
**Figure supplement 1.** Role of CaaX motif in OAS1 localization.

(p46ATIL) caused a significant loss in EMCV RNA binding (*Figure 4B*). These data confirm that endo-membrane targeting through the CaaX motif is required for p46 to bind EMCV RNA, as mutating the CaaX motif completely ablated the ability of p46 to access EMCV RNA (*Figure 4B*).

To investigate the role of the CaaX motif to the antiviral activity of OAS1, we expressed p42, p46, p42CTIL, or p46ATIL in *OAS1* KO 293 T cells followed by infection with EMCV (*Figure 4C*). Compared to cells expressing p46, cells expressing CaaX-mutant p46ATIL showed a significant five-fold increase in EMCV RNA as well as an increase in viral titer; a decreased antiviral activity similar to that of the p42 isoform (*Figure 4D and E*). Intriguingly, addition of a CaaX motif to OAS1 p42 did not increase the antiviral activity of p42 (*Figure 4D and E*). One possible explanation for this discrepancy could be that the peptide sequence in the unique C-terminus of p42 contains a motif that is inhibitory to its antiviral activity. We therefore generated another OAS1 variant which contains a CaaX motif directly downstream of the common OAS1 peptide sequence shared between all OAS1 isoforms, hereinafter referred to as OAS1 common+CTIL (*Figure 4F*). We tested the antiviral activity of OAS1 common+CTIL compared to the p42 and p46 isoforms. At similar protein expression levels, OAS1 common+CTIL did not confer significantly increased antiviral activity over the empty vector control and showed RNA levels similar to those in cells expressing the p42 isoform (*Figure 4G and H*). Interestingly, the OAS1 common+CTIL variant showed significantly decreased co-localization with Golgi membranes compared to p46 and appeared cytosolic and nuclear, as observed above for CaaX-deficient p42 (*Figure 4—figure supplement 1B*). Together, these data indicate membrane targeting of OAS1 p46 through a CaaX motif is crucial for its antiviral activity against EMCV. However, a CaaX motif alone is not sufficient to provide optimal antiviral activity against EMCV. Overall, these data suggest other features in the unique C-termini of OAS1 p42 and p46 contribute to their individual antiviral activity.

## Combined effects of CaaX motif, C-terminus length and oligomerization domain confer differential antiviral activity of OAS1 isoforms

Since the unique C-terminus of p46 (50 aa) is longer than the unique C-terminus of p42 (18 aa), we investigated if the unique sequence in p46 contains additional motifs important for its antiviral activity. We generated sequential hexameric alanine substitution mutants throughout the C-terminus of OAS1 p46 and compared their antiviral activity against EMCV alongside p46 and CaaX-deficient p46ATIL (*Figure 5A*). As expected from our previous experiments, p46ATIL had reduced antiviral activity when compared with p46 (*Figure 5B and C*, and *Figure 5—figure supplement 1A*). However, compared to p46, none of the alanine substitution mutants affected viral RNA levels significantly (*Figure 5C* and *Figure 5—figure supplement 1A*). Notably, mutant eight showed a trend toward increased EMCV RNA levels, suggesting that the mutated amino acids might contribute to the antiviral activity of OAS1 p46. Although OAS1 p46 mutant eight localized to the Golgi, compared with p46, this mutant showed a slight but significant decrease in its co-localization with Golgin-97 (*Figure 5—figure supplement 1B and C*). In fact, the $E^{392}$-E/$N^{393}$-D/$N^{394}$ sequence is conserved in most mammalian OAS1 p46 orthologous isoforms, which suggests a possible function

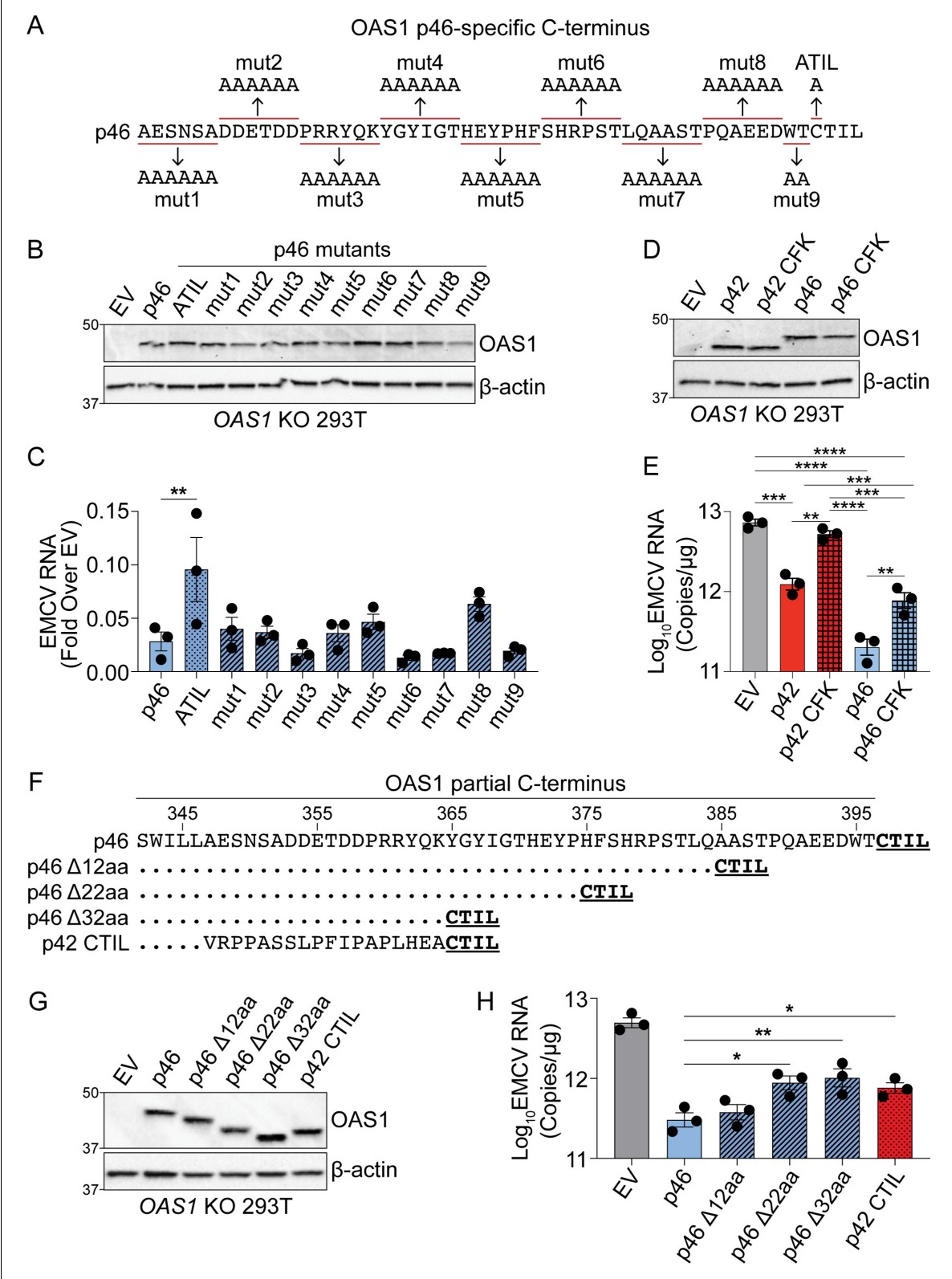

**Figure 5.** Combined effects of CaaX motif, C-terminus length and oligomerization domain confer differential antiviral activity of OAS1 isoforms. (**A**) C-termini of OAS1 p46 alanine substitution mutants 1–9. (**B**) Immunoblot of control EV, p46, p46ATIL, and p46 alanine substitution mutants 1–9 in *OAS1* KO 293 T cells at 24 hr post transfection. (**C**) Quantification of EMCV 5′UTR by RT-qPCR from *OAS1* KO 293 T cells expressing a control EV, p46, p46ATIL, and p46 alanine substitution mutants 1–9 at 24 hr post EMCV infection (MOI=0.001). (**D**) Immunoblot of control EV, p42, p42 CFK mutant, p46,

*Figure 5 continued on next page*

*Figure 5 continued*

and p46 CFK mutant in *OAS1* KO 293 T cells at 24 hr post transfection. (E) Quantification of EMCV 5′UTR by RT-qPCR from *OAS1* KO 293 T cells expressing a control EV, p42, p42 CFK mutant, p46, and p46 CFK mutant at 24 hr post EMCV infection (MOI=0.001). (F) Alignment of C-termini of OAS1 p46, p46 truncation mutants Δ12aa, Δ22aa, and Δ32aa, and p42CTIL. (G) Immunoblot of control EV, p46, p46 truncation mutants Δ12aa, Δ22aa, and Δ32aa, and p42CTIL in *OAS1* KO 293 T cells at 24 hr post transfection. (H) Quantification of EMCV 5′UTR by RT-qPCR from *OAS1* KO 293 T cells expressing EV, p46, p46 truncation mutants Δ12aa, Δ22aa, and Δ32aa, and p42CTIL at 24 hr post EMCV infection (MOI=0.001). (C) and (H) Data were analyzed by one-way ANOVA with Dunnett's multiple comparisons test (vs. p46) where *$p<0.05$, and **$p<0.01$. (D). For (E) data were analyzed using one-way ANOVA with Tukey's multiple comparisons test where **$p<0.01$, and ***$p<0.001$ and ****$p<0.001$. (B, D) and (G) Representative immunoblots of three independently performed experiments are shown. (C, E) and (H) Each data point represents an independently performed experiment. The online version of this article includes the following source data and figure supplement(s) for figure 5:

**Source data 1.** Uncropped gels for the associated panels in *Figure 5* and *Figure 5—figure suppleent 1*.
**Figure supplement 1.** Enhanced antiviral activity of OAS1 p46 is mediated by p46-specific coding region.

for this motif (*Figure 1—figure supplement 1C*). Nevertheless, none of the alanine substitution mutants affected EMCV RNA levels as strongly as the disruption of the CaaX motif in p46ATIL.

Previous studies have demonstrated a three amino acid motif ($C^{331}F^{332}K^{333}$), present in all OAS1 isoforms, is critical for OAS1 oligomerization and 2′−5′A production (*Ghosh et al., 1997*). The CFK motif is in close proximity to the unique C-termini of p42 and p46, however, the requirement of the CFK domain for the antiviral activity of these proteins is unknown. Therefore, we generated OAS1 p42 and p46 CFK mutants (C331A-F332A-K333A) and compared their antiviral activity against EMCV to their wild type counterparts. Relative to wild type p42 and p46, mutation of the CFK domain led to a significant loss in the ability of p42 and p46 to reduce EMCV RNA by threefold and twofold, respectively (*Figure 5D and E*). Interestingly, disruption of the CFK motif in p42 resulted in complete loss of antiviral activity, demonstrated by similar EMCV RNA copies as cells expressing an empty vector control (*Figure 5D and E*). OAS1 p46 with a disrupted CFK motif maintained residual antiviral activity similar to the antiviral activity of wild-type p42 (*Figure 5D and E*). These data show that the antiviral activity of OAS1 p42 and p46 is partially mediated by CFK motif-dependent oligomerization. This oligomerization facilitates enhanced 2′−5′A synthesis capacity (*Figure 5—figure supplement 1D*). Furthermore, we found that CFK-mediated oligomerization is required for optimal EMCV RNA binding of OAS1 p46 (*Figure 5—figure supplement 1E*). Importantly, mutation of the CFK motif did not alter the localization of either isoform (*Figure 5—figure supplement 1F and G*). This suggests that the antiviral activity of OAS1 p42 is completely dependent on the CFK oligomerization motif, whereas p46 only partially depends on the CFK oligomerization motif for its antiviral activity.

Since p46 is prenylated and embedded in cellular membranes, the longer C-terminus of p46 may function as a flexible linker to facilitate optimal oligomerization. This may explain why adding a CaaX motif to p42 is not sufficient to enhance its antiviral activity (see *Figure 4C–E*). To test this, we generated several C-terminal truncation mutants of p46 termed p46 Δ12aa, Δ22aa, and Δ32aa, and determined their antiviral activity against EMCV alongside p46 and p42CTIL (*Figure 5F*). Compared to p46, the truncation mutants showed a length-dependent decrease in their ability to reduce EMCV RNA (*Figure 5G and H*). The Δ22aa and Δ32aa deletion mutants had similar antiviral activities as p42CTIL, which is identical in length to the Δ32aa mutant (*Figure 5H*). Similar to p46, all the p46 truncation mutants localized to the Golgi (*Figure 5—figure supplement 1H*). However, there was a slight but significant decrease in the correlation of p46 Δ32aa with the Golgi (*Figure 5—figure supplement 1I*). These data suggest that, in addition to the CaaX motif and an oligomerization domain, the C-terminal length of p46 is important for its antiviral activity.

Furthermore, we observed the conservation of a longer CaaX-containing p46-specific C-terminal sequence without sequence similarity in various other species. Interestingly, the p46-specific C-terminal sequence was truncated in rodents and bats. To test if the length of the p46-specific C-terminal sequence affected the antiviral activity against EMCV, we created chimeric proteins by inserting the C-terminal sequences of different lengths specific to the p46 orthologs of cow (*Bos taurus*), fox (*Vulpes vulpes*), flying fox (*Pteropus alecto*), and alligator (*Alligator mississippiensis*) into the human OAS1 p46 gene (*Figure 5—figure supplement 1J*). We selected species with divergent OAS1 p46-specific C-terminal sequences and minimal amino acid sequence identities when compared to human OAS1 p46. We found cow and fox OAS1 chimeras mimicked human OAS1 p46 antiviral activity, whereas the shorter C-terminus of flying fox led to decreased antiviral activity of the chimeric protein

compared to human OAS1 p46 (*Figure 5—figure supplement 1J*). These data are consistent with our hypothesis that the length of the p46-specific C-terminal sequence is important for increased antiviral activity. Intriguingly, the alligator OAS1 p46 chimeric protein, with similar C-terminus length as human p46 but with most divergence in peptide sequence, failed to exhibit similar antiviral activity as human OAS1 p46 (*Figure 5—figure supplement 1J*). Upon close examination, we found that the alligator (and guinea pig) lack conservation of the $E^{392}$-E/$N^{393}$-D/$N^{394}$ motif proximal to the CaaX motif, which we have shown is required for strong antiviral activity (*Figure 5A–C* and *Figure 1—figure supplement 1C*). These data suggest that the length of the p46-specific C-terminal sequence, CFK oligomerization domain, and potentially E-E/N-D/N sequence are required for enhanced antiviral activity of OAS1 p46.

## OAS1 p46 has broad antiviral activity against viruses that use the endomembrane system for replication

We sought to define the antiviral activity of OAS1 isoforms against other positive-strand RNA viruses that use the endomembrane system for replication. We tested if OAS1 isoforms are differentially antiviral against West Nile virus (WNV) by infecting *OAS1* KO 293 T cells expressing p42, p46, or control with WNV. WNV, like all flaviviruses, replicates on membranes of the endomembrane system, primarily ER-derived membranes (*Westaway et al., 1997*). Expression of p46 led to a 90% reduction in WNV titer relative to control, while p42 did not significantly impact WNV titer (*Figure 6A*). These data demonstrate that OAS1 p46 has enhanced antiviral activity against WNV.

Next, we tested if the enhanced antiviral activity of p46 against WNV could be explained by p46 localizing to sites of flavivirus RNA replication. We infected *OAS1* KO 293 T cells expressing p42 and p46 with WNV, followed by staining for OAS1, dsRNA, and the endoplasmic reticulum protein disulfide-isomerase A3 (PDIA3), since WNV replicates within invaginations of the ER membrane. During WNV infection, we observed p42 in the cytosol and nucleus. Indeed, p42 did not appear to be recruited to sites of WNV RNA production (*Figure 6B*, top panel, arrows). However, during WNV infection, p46 localized to PDIA3 and dsRNA-positive replication sites (*Figure 6B*, bottom panel, arrows). Quantification of OAS1 isoforms relative to PDIA3 revealed a significant increase in the correlation of p46, but not p42, to PDIA3-positive membranes during infection, suggesting that p46 is recruited to sites of WNV replication (*Figure 6B and C*). Quantification of OAS1 isoform localization to viral RNA revealed OAS1 p46 has significantly stronger localization to dsRNA during WNV infection than p42 (*Figure 6D*).

We evaluated whether p46 was present in the Golgi during WNV infection by staining for OAS1, Golgin-97, and dsRNA in cells infected with WNV. OAS1 p46 was significantly more correlated with the Golgi than p42 in both uninfected and WNV-infected cells. However, WNV infection caused a significant decrease in the association of p46 with Golgin-97 (*Figure 6—figure supplement 1A and B*). This suggests that p46 may be recruited from the Golgi to sites of WNV replication and is consistent with previous observations that Golgi membranes and proteins, including RNA binding proteins, are recruited to the replication organelles of flaviviruses (*Ward et al., 2016*). Alternatively, p46 might be recruited during prenylation at the ER membrane.

We next tested if p46 localizes to sites of Zika virus (ZIKV) replication, another flavivirus that utilizes ER membranes for replication (*Cortese et al., 2017*). We observed p46, but not p42, localized to ZIKV replication sites at the ER (*Figure 6—figure supplement 1C*). Overall, these data suggest that endomembrane localization of OAS1 p46 allows this protein to access sites of flavivirus RNA replication.

The human pathogenic picornavirus coxsackievirus B3 (CVB3) replicates within modified Golgi membranes, and thus p46 may also be positioned to readily sense CVB3 RNA (*Melia et al., 2019*). We tested the antiviral activity of OAS1 p42 and p46 against CVB3 by infecting *OAS1* KO 293 T cells expressing p42, p46 or a control. RT-qPCR analysis at 24 hr post-infection revealed a 70% reduction in CVB3 RNA in p46 expressing cells compared to control, while p42 had almost no effect on CVB3 RNA levels (*Figure 6—figure supplement 1D*). Similarly, CVB3 titers at 48 hr post-infection were significantly reduced by 50% in p46 expressing cells (*Figure 6E*). These data suggest that OAS1 p46 may have broad antiviral activity against members of the picornavirus family, while p42 is less effective.

Coronaviruses such as SARS-CoV-2 use the endomembrane system for replication using primarily ER-derived membranes (*Romero-Brey and Bartenschlager, 2014*; *Snijder et al., 2020*). We

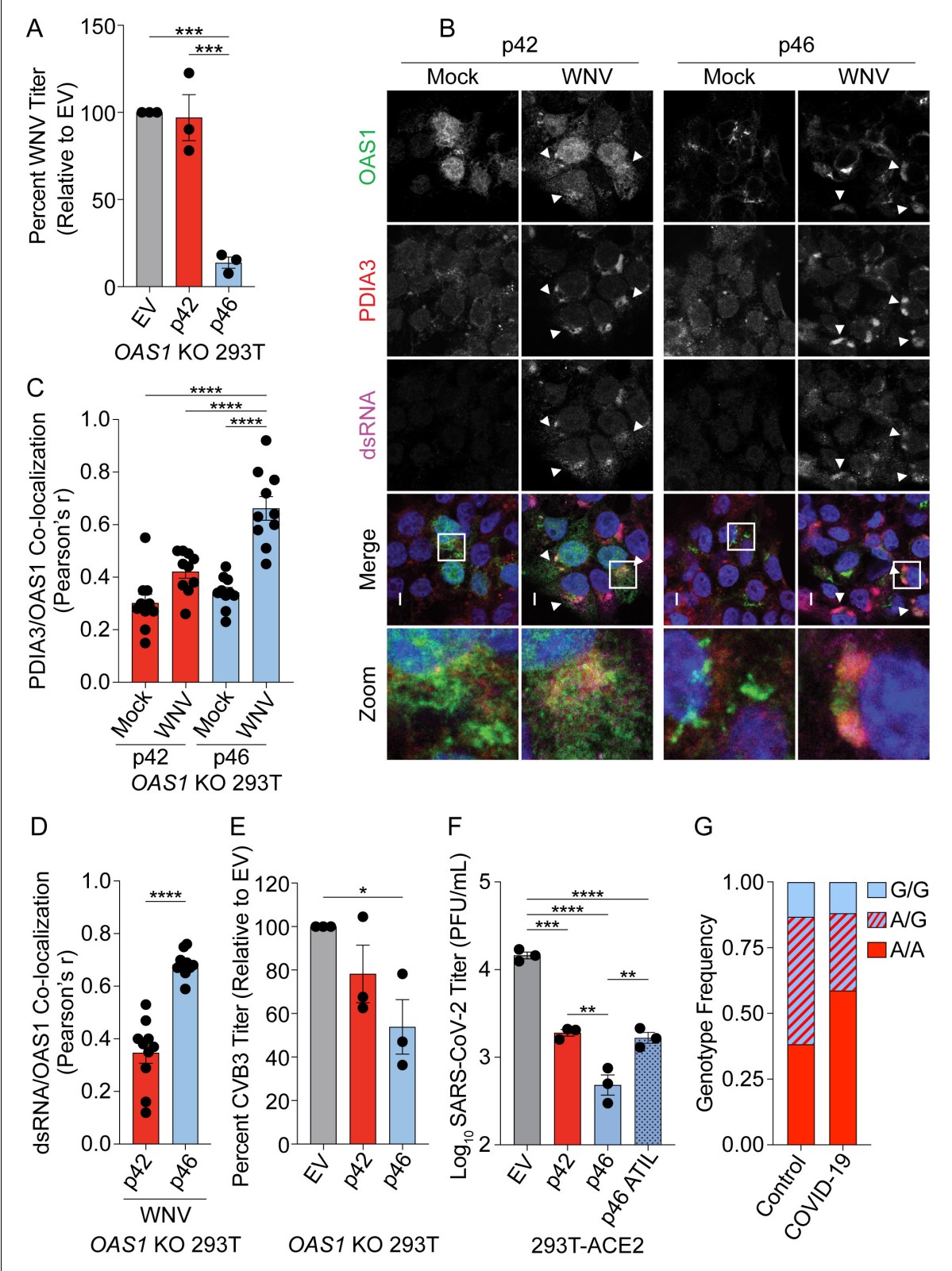

**Figure 6.** OAS1 p46 has broad antiviral activity against viruses that use the endomembrane system for replication. (**A**) WNV Texas titers (percent titer normalized to EV) 48 hr post WNV infection (MOI=0.001) taken from *OAS1* KO 293 T cells transfected with control EV, p42, and p46. (**B**) Representative confocal micrographs from mock or WNV Texas (MOI=1, 24 hr) infected *OAS1* KO 293 T cells expressing p42 or p46 and stained with DAPI (blue) and anti-OAS1 (green), PDIA3 (red), and dsRNA (magenta) antibodies; scale = 5 μm. (**C, D**) Pearson's correlation of OAS1 and (**C**) PDIA3 and (**D**) dsRNA;

*Figure 6 continued on next page*

*Figure 6 continued*

each data point represents an individual cell of one representative experiment. (**E**) CVB3 titers (percent titer normalized to EV) 48 hr post CVB3 infection (MOI=0.001) in *OAS1* KO 293 T cells transfected with a control EV, p42, or p46. (**F**) SARS-CoV-2 titers taken at 48 hr post-infection (MOI=0.1) from ACE2 293 T cells expressing EV, p42, p46, or p46ATIL for 24 hr. (**G**) *OAS1* rs10774671 genotype frequency in severe COVID-19 cohort and matched healthy control subjects. (**A**) (**C**) (**E**) and (**F**) Data were analyzed using one-way ANOVA with Tukey's multiple comparisons test where *p<0.05, **p<0.01, ***p<0.001, ****p<000.1 (**B**) Representative cells from one of two independently performed experiments are depicted. (**D**) Data were analyzed by unpaired t test where ****p<0.0001. (**A**), (**E**) and (**F**) Each data point represents an independently performed experiment.

The online version of this article includes the following source data and figure supplement(s) for figure 6:

**Source data 1.** Excel file with the summary statistics for the association analysis for *Figure 6G*.
**Figure supplement 1.** Endomembrane targeting of OAS1 p46 confers its enhanced antiviral activity.

---

therefore hypothesized that p46 may have enhanced antiviral activity against coronaviruses. We assessed the antiviral activity of p42 and p46 against SARS-CoV-2 in 293 T cells expressing the SARS-CoV-2 entry receptor ACE2. Expression of p42 led to a fivefold reduction in SARS-CoV-2 titer (*Figure 6F*). However, p46 had a significant fivefold greater antiviral activity over p42 (25-fold over EV) against SARS-CoV-2 (*Figure 6F*). Importantly, the enhanced antiviral activity of p46 against SARS-CoV-2 depended on the CaaX motif, as p46ATIL demonstrated decreased antiviral activity similar to that of p42. This suggests that endomembrane targeting of p46 is critical to its enhanced antiviral activity against SARS-CoV-2 (*Figure 6F*).

We next tested the antiviral activity of p42 and p46 against negative-strand RNA viruses, such as Influenza A virus (IAV) and Indiana vesiculovirus (formerly VSV), which do not replicate on intracellular organelle membranes. IAV replicates within the nucleus where p42 is also present and is sensitive to the OAS/RNase L pathway (*Li et al., 2016*; *Min and Krug, 2006*). To test if OAS1 isoforms are differentially antiviral against IAV, we infected *OAS1* KO 293 T cells expressing p42, p46, or a control with influenza A/PR/8/34 (*Figure 6—figure supplement 1E*). Viral RNA analysis revealed neither OAS1 isoform impacted IAV RNA levels significantly at 24 hr post-infection. Mutations in IAV NS1 disrupting RNA binding render this virus sensitive to the OAS/RNase L pathway, although it is unclear what antiviral role OAS1 isoforms play during IAV infection (*Min and Krug, 2006*). We found that expression of p42 or p46 had no impact on the replication of this NS1-mutant IAV (*Figure 6—figure supplement 1F*). Similarly, we observed that GFP-expressing VSV, which replicates in the cytosol, was also insensitive to OAS1 p42 and p46. Compared to an empty vector control, expression of p42 or p46 did not impact the number of GFP+ cells during infection (*Figure 6—figure supplement 1G*). Collectively, these data suggest OAS1 p46 is broadly antiviral against viruses that use the endomembrane system for replication.

### *OAS1* rs10774671 is associated with severe COVID-19 disease

Since the *OAS1* rs10774671 A/G variant generates the OAS1 p42 and p46 isoforms, respectively, that affected SARS-CoV-2 titers in vitro, we tested whether this SNP is associated with COVID-19 disease severity. We hypothesized that the G allele that encodes the p46 isoform would decrease the risk of severe COVID-19 complications due to its superior antiviral activity compared to the A allele that generates p42. To test this, we genotyped the rs10774671 SNP in 34 COVID-19 severe cases (hospitalized, requiring mechanical ventilation) and 99 ancestry matched healthy controls (*Supplementary file 5*). Association was tested by logistic regression, adjusting for sex and self-reported ancestry. We detected association of rs10774671 in severe COVID-19 cases (p=0.017, Odds Ratio 0.35, 95% CI 0.15–0.83) using a dominant model, indicating that the G allele was protective for severe COVID-19 disease (*Supplementary file 5* and *Figure 6G*).

These data are consistent with results from a recent genome wide association study of 1676 critically ill COVID-19 patients of European descent and UK Biobank controls (n=8,380) which identified a significant association with the *OAS1/OAS2/OAS3* locus (*Pairo-Castineira et al., 2021*). The lead SNP in this region, rs10735079 (p=$1.65 \times 10^{-8}$, OR 1.3, 95% CI 1.18–1.42), is in high linkage disequilibrium with the rs10774671 A/G splicing site variant (D'0.91, $r^2$=0.79). To replicate our findings, we tested association of rs10774671 in these cohorts by logistic regression, correcting for age, sex, postal code deprivation decile, and principal components of ancestry. Significant association of the rs10774671 G allele with severe COVID-19 was detected (p=$7.38 \times 10^{-7}$, OR 0.80 95% CI 0.71–0.89).

Together these results demonstrate that the G allele at *OAS1* rs10774671 encoding the OAS1 p46 isoform contributes protection from severe disease in SARS-CoV-2-infected patients.

## Discussion

In this study, we show that endomembrane targeting of OAS1 p46 confers enhanced antiviral activity against viruses that use the endomembrane system for their replication. The p46 isoform contains a functional CaaX motif that targets this isoform to the endomembrane system, primarily to Golgi membranes, while other OAS1 isoforms localize to the cytosol and nucleus. Compared with p42, the p46 isoform has enhanced antiviral activity against picornaviruses, flaviviruses, and SARS-CoV-2, all viruses that replicate their RNA within modified organelles of the endomembrane system (*Cortese et al., 2017*; *Melia et al., 2019*; *Melia et al., 2018*; *Snijder et al., 2020*). Although replicating in these modified endomembrane compartments is an important immune evasion strategy of positive-strand RNA viruses, out data shows localizing OAS1 p46 to the endomembrane system results in enhanced access to viral RNA and activation of RNase L dependent antiviral activity. In contrast, p42, which is localized to the cytosol and nucleus, has a comparatively weak ability to sense viral RNA and initiate antiviral activity against viruses that replicate within intracellular membranes (*Figure 7*).

RNA replication for positive-strand RNA viruses occurs within the replication organelle. Placement of innate immune sensors at intracellular membranes is an antiviral strategy likely difficult for positive-strand RNA viruses to evade, as these viruses are unlikely to evolve away from this fundamental replication strategy. Based on our data, we propose that prenylated OAS1 p46 localizes to the cytosolic surface of the organelle membrane. During infection with a positive-strand RNA virus, the viral non-structural proteins and viral RNA are recruited to the organelle surface, where p46 can interact with vRNA. As the membrane is involuted to form the viral replication organelle (VRO), prenylated p46 is also included in the VRO. Confinement of p46 in the VRO may enhance its access to viral RNA. Furthermore, newly prenylated proteins are continuously recruited to the organelle surface during infection, which may allow for additional interference of vRNA as it escapes the replication organelle.

Secondly, many positive-strand RNA viruses depend on lipid products of the mevalonate pathway or prenylation for replication and therefore cannot broadly antagonize this pathway (*Mackenzie et al., 2007*; *Romero-Brey and Bartenschlager, 2014*). Viral antagonism by specifically targeting OAS1 p46 protein or directly blocking its prenylation or splicing could be possible. Alternatively, viral interference of the OAS-RNase L pathway would negate the antiviral activity OAS1 p46. Such a mechanism is exemplified in the case of some viruses, where the non-structural proteins inhibit the OAS-RNase L pathway (*Silverman, 2007*; *Thornbrough et al., 2016*; *Zhao et al., 2012*). Among positive sense RNA viruses Mouse Hepatitis Virus (MHV) and MERS-CoV accessory protein encodes for a phosphodiesterase which degrade 2′−5′A. As 2′−5′A is critical for the activation of RNAseL, antagonizing this pathway makes them insensitive to OAS1-RNaseL-mediated virus control. However, intriguingly, SARS-CoV-2 do not carry this phosphodiesterase in their genome, probably making them susceptible to OAS1 p46.

We also show addition of the CaaX motif to p42 is not sufficient to enhance its ability to bind viral RNA or increase its antiviral activity. This suggests that the CaaX motif is just one of the crucial features in the unique OAS1 p46 C-terminal region that contributes to differential antiviral activity. Tetramerization of OAS1 has been proposed as a requirement for synthesis of the second messenger 2′−5′A, indicating that several OAS1 molecules in close proximity are required to induce 2′−5′A synthesis upon viral RNA binding and to subsequently activate the RNase L pathway (*Ghosh et al., 1997*). Aggregation of several OAS1 proteins around viral dsRNA might also increase the number of RNA-binding domains within the OAS1/vRNA complex, thereby allowing access to longer vRNA which would increase 2′−5′A synthesis. Mutation of the CFK oligomerization motif in OAS1 p42 led to a complete loss of antiviral activity, while the p46 CFK mutant maintained antiviral activity similar to that of the wildtype p42 isoform. This suggests a model in which OAS1 p46 without a functional CFK oligomerization domain still maintains a residual antiviral activity. These data show p46 proteins might oligomerize which is further aided by CaaX motif-mediated localization at Golgi membranes. Together, we have defined features including the CaaX motif in the unique C-terminus that contribute to the enhanced antiviral activity of p46. The unique alternatively spliced C-terminal region

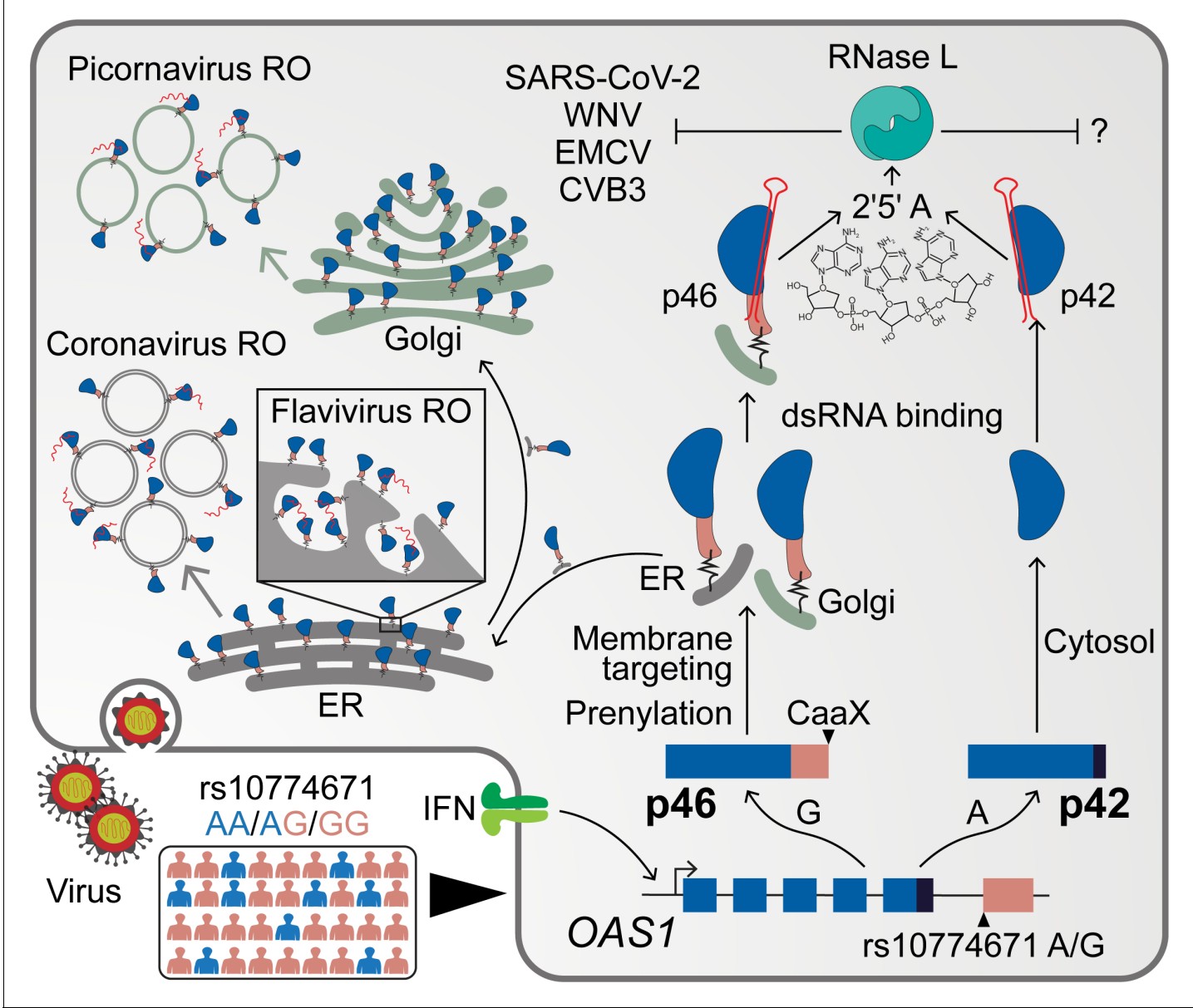

**Figure 7.** Schematic depicting how endomembrane targeting of OAS1 p46 primes antiviral activity against positive-strand RNA viruses. A splice-acceptor SNP (rs10774671) controls production of the p42/p46 OAS1 isoforms. Isoform-specific prenylation localizes p46 to the Golgi apparatus, while OAS1 p42 is cytosolic. During positive-strand RNA virus infection, OAS1 p46 is recruited to virus replication organelles (VROs) of flaviviruses, picornaviruses and coronaviruses. Through this targeting p46 gains enhanced access to viral RNA. OAS1 p42 remains cytosolic and nuclear during infection and has limited access to viral RNA. Both OAS1 isoforms require catalytic activity and RNase L to be antiviral.

including the oligomerization and prenylation motifs in p46 is maintained across vertebrates presumably because it displays the strongest antiviral activity compared to other OAS1 isoforms (*Figure 7*).

The rs10774671 G allele controlling OAS1 p46 expression was originally identified as a West Nile virus resistance allele, although the mechanism accounting for the protection conferred by this SNP was lacking (*Lim et al., 2009*). In this study, we identify a functional *OAS1* splice site variant affects COVID-19 outcome. We found the A allele contributes to genetic risk for severe COVID-19 disease in patients with respiratory failure compared to healthy control subjects in two cohorts. We propose that the G allele association with protection against severe COVID-19 is explained by the enhanced antiviral activity of OAS1 p46 against SARS-CoV-2, which contributes to early control of viral replication. Additionally, both rs10774671 and the lead GWAS SNP rs10735079 are expression and splicing

QTL for *OAS1* and *OAS3* in our analyses (*Pairo-Castineira et al., 2021*). Thus, the association of these SNPs with severe COVID-19 may be two-fold: affecting splicing and expression levels both of those observed in primary human cells.

We have shown that the splice-site SNP controls the subcellular targeting of a critical viral RNA sensor, OAS1, which may allow the host to respond to viral evolutionary pressures and replication strategies. Interestingly, the G allele is ancestral and prevalent in African populations. In contrast, the A allele is prevalent in rest of the world (*Figure 1—figure supplement 1B*). Previous studies have documented reintroduction of the G-allele into *Homo sapiens* population by introgression from Neanderthals (*Sams et al., 2016*), and this could have implications for COVID-19 disease. Other innate immune genes upstream of OAS1 that are associated with severe COVID-19 in a genome wide association study are *IFNAR2*, and *TYK2*, which are in the type I IFN pathway (*Pairo-Castineira et al., 2021*). Perturbations in any of these genes could affect OAS1 expression and its effector activity. Genetic studies, SARS-CoV-2 antagonism of IFN transcription, and autoantibodies against IFNα/β strongly associate with severe COVID-19, indicating that the type I IFN pathway might be compromised in patients with severe COVID-19 (*Bastard et al., 2021*; *de Prost et al., 2021*; *Pairo-Castineira et al., 2021*; *Xia et al., 2020*). While it would be interesting to study the individual and combined effects of the above genes on OAS1 and COVID-19 disease, our study suggests there are strong selective pressures, presumably viral, impacting the prevalence of OAS1 causal splice site-SNP in in human populations. (*Li et al., 2017*; *Liu et al., 2017*; *O'Brien et al., 2010*).

The broad number of RNA- and even DNA-viruses inhibited by the OAS/RNAse L pathway has raised interesting questions about the determinates of OAS antiviral specificity (*Silverman, 2007*). In humans, the OAS family is comprised of three catalytically active OAS proteins: OAS1, OAS2, and OAS3. OAS1 contains an RNA binding domain and a catalytically active domain, while OAS2 contains a catalytically inactive repeat of this unit, and OAS3 contains two catalytically inactive repeats of this unit (*Hornung et al., 2014*). The effect of OAS3 SNPs that associate with COVID-19 is still unclear. Although the OAS proteins have different RNA binding capabilities and favor synthesis of different lengths of $2'-5'$A (*Ibsen et al., 2014*), we demonstrate a novel mechanism where subcellular localization determines the antiviral specificity of these proteins. Based on the reduction of vRNA in our study, we propose a model in which $2'-5'$A locally activates RNase L around sites of viral replication. While the function of OAS1 p42 is unclear but might confer resistance to yet unknown pathogen. This is further supported by the genetic association of the A allele with multiple autoimmune disorders, as enhanced immune responses are sometimes made at a tradeoff for overall host fitness (*Li et al., 2017*; *Liu et al., 2017*; *O'Brien et al., 2010*).

Our work highlights intracellular targeting as a crucial determinant for the specificity of OAS1. By positioning viral RNA sensors at different subcellular sites of viral RNA accumulation, the host can potentially respond to diverse replication strategies in the cytosol, nucleus, or on intracellular membranes. Although this work focused on OAS1 and the detection of viral nucleic acids, subcellular targeting is likely also important for function of other OAS proteins. Future studies on the subcellular localization of other OAS proteins will define how this pathway is able to respond to viruses with diverse intracellular replication strategies.

## Acknowledgements

This project was funded by National Institutes of Health grants (nos. AI145974, AI108765, AI135437 to RS; AI104002, AI118916, AI145296, AI127463, AI100625 to MG); T32 and F31 training grants (nos. AI106677, GM007270, and AI140530 to FWS; T32 HL007312 to AFR); a Postdoctoral Research Fellowship from the German Research Foundation (JS); JMC received support from the Cancer Research Institute Irvington Fellowship Program. We thank the BRI COVID-19 Research Team for collective the samples for genetic analysis. We thank M A Davis (UW Immunology) for help with confocal laser scanning microscopy. The Sapphire Biomolecular Imager (Azure Biosystems) used for this work was supported by the Office of the Director of the National Institutes of Health under award S10OD026741.

## Additional information

### Funding

| Funder | Grant reference number | Author |
| --- | --- | --- |
| National Institutes of Health | AI145974 | Ram Savan |
| National Institutes of Health | AI108765 | Ram Savan |
| National Institutes of Health | AI135437 | Ram Savan |
| National Institutes of Health | AI104002 | Michael Gale Jr |
| National Institutes of Health | AI118916 | Michael Gale Jr |
| National Institutes of Health | AI145296 | Michael Gale Jr |
| National Institutes of Health | AI127463 | Michael Gale Jr |
| National Institutes of Health | AI100625 | Michael Gale Jr |
| National Institutes of Health | AI106677 | Frank W Soveg |
| National Institutes of Health | GM007270 | Frank W Soveg |
| National Institutes of Health | AI140530 | Frank W Soveg |
| National Institutes of Health | T32 HL007312 | Adriana Forero |
| NIH Office of the Director | S10OD026741 | Joshua J Woodward |
| German Research Foundation | | Johannes Schwerk |
| Cancer Research Institute | | Jonathan M Clingan |

The funders had no role in study design, data collection and interpretation, or the decision to submit the work for publication.

### Author contributions

Frank W Soveg, Johannes Schwerk, Conceptualization, Data curation, Formal analysis, Investigation, Methodology, Writing - original draft, Writing - review and editing; Nandan S Gokhale, Karen Cerosaletti, Data curation, Formal analysis, Investigation, Methodology, Writing - original draft, Writing - review and editing; Julian R Smith, Adriana Forero, Shivam A Zaver, Data curation, Formal analysis, Investigation; Erola Pairo-Castineira, Cate Speake, Data curation, Formal analysis; Alison M Kell, Katharina Esser-Nobis, Justin A Roby, Investigation, Methodology; Tien-Ying Hsiang, Saumendra N Sarkar, Daniel B Stetson, Jane H Buckner, Resources, Methodology; Snehal Ozarkar, Data curation, Investigation, Methodology; Jonathan M Clingan, Eric J Allenspach, Resources; Eileen T McAnarney, Chiraag Balu, Investigation; Amy EL Stone, Methodology; Uma Malhotra, Joseph Perez, Jennifer L Hyde, Joshua J Woodward, Resources, Investigation, Methodology; Vineet D Menachery, Resources, Investigation; John Kenneth Baillie, Resources, Data curation, Formal analysis, Investigation; Michael Gale Jr, Resources, Funding acquisition, Investigation, Methodology; Ram Savan, Conceptualization, Resources, Supervision, Investigation, Writing - original draft, Project administration, Writing - review and editing

### Author ORCIDs

Frank W Soveg (ID) https://orcid.org/0000-0002-3403-0190
Johannes Schwerk (ID) https://orcid.org/0000-0002-5964-7162
Karen Cerosaletti (ID) http://orcid.org/0000-0002-7403-6239
Adriana Forero (ID) http://orcid.org/0000-0002-2698-658X
Katharina Esser-Nobis (ID) http://orcid.org/0000-0002-9027-0391
Jonathan M Clingan (ID) http://orcid.org/0000-0003-4729-0247
Eileen T McAnarney (ID) http://orcid.org/0000-0003-2337-2889
Amy EL Stone (ID) https://orcid.org/0000-0002-9140-6510
Cate Speake (ID) http://orcid.org/0000-0003-1480-4272
Eric J Allenspach (ID) https://orcid.org/0000-0001-7346-5835

Jennifer L Hyde (ID) https://orcid.org/0000-0001-8062-1672
Saumendra N Sarkar (ID) https://orcid.org/0000-0002-2850-6121
Joshua J Woodward (ID) http://orcid.org/0000-0002-4630-403X
Daniel B Stetson (ID) http://orcid.org/0000-0002-5936-1113
John Kenneth Baillie (ID) https://orcid.org/0000-0001-5258-793X
Ram Savan (ID) https://orcid.org/0000-0002-3087-1355

## Ethics

Human subjects: A cohort of 99 healthy control subjects matched for ancestry (self-reported) was assembled from participants in the 1310 healthy control registry at Benaroya Research Institute. Both studies were approved by the Institutional Review Board at Benaroya Research Institute (IRB20-036 and IRB07109 respectively).

## Decision letter and Author response

Decision letter https://doi.org/10.7554/eLife.71047.sa1
Author response https://doi.org/10.7554/eLife.71047.sa2

## Additional files

### Supplementary files

• Source data 1. Cumulative PDF file of all the source data in the manuscript.

• Supplementary file 1. Cells used in this study.

• Supplementary file 2. Antibodies used in this study.

• Supplementary file 3. Nucleic acids used in this study.

• Supplementary file 4. Viruses used in this study.

• Supplementary file 5. Clinical and demographic information for COVID-19 and matched healthy control cohort.

• Transparent reporting form

### Data availability

All data generated during this study are provided in the manuscript, supporting files, and source data files. Raw PLINK results for the association analysis of local subjects are provided in Supplementary file 5; association data for the GenOMICC replication cohort is available as described in the primary publication (Pairo-Castineira et al., 2020). Additional data on our local COVID-19 cohort is available upon request (KCerosaletti@benaroyaresearch.org). This is being done to protect the privacy of the subjects in this study as the data were obtained from samples recovered from the hospital clinical laboratory with IRB approval but without written consent. For commercial entities, availability of these data will be assessed on a case-by-case basis in conjunction with the Benaroya Research Institute business development office.

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
