## [Decision Letter]

[Editors' note: this paper was reviewed by Review Commons.]

**Acceptance summary:**

The manuscript of Soveg et al. provides an analysis of isoforms of the OAS1 protein and their role in viral infection. The authors convincingly demonstrate that the p46 isoform of OAS1 is prenylated and targeted to intracellular membranes. The molecule has specific antiviral activity against several viruses. In a small association study, the authors provide results that support published data for the genetic impact of this region on SARS-CoV-2 severity.

---

## [Author Response]

Reviewer #1 (Evidence, reproducibility and clarity (Required)):The manuscript of Soveg et al. describes an analysis of isoforms of the OAS1 gene and their role in viral infection. The authors also show convincing evidence that the p46 isoform of OAS1 is prenylated and targeted to intracellular membranes and has specific antiviral activity against several viruses. The authors also present a small association study that supports published data for the genetic impact of this region on SARS-CoV-2 severity. The paper is well written, and the data support the conclusions.

We thank this reviewer for the assessment of our study with helpful suggestions to improve on the manuscript. We have addressed your concerns in our point-by-point response below:

The paper could be improved as follows:Major comments:The conclusions are convincing and there are no highly speculative claims.Is there genetic evidence for the role of this variant in other viruses implicated here such WNV?

Yes, the reviewer is correct, in the introduction, we have cited studies that show genetic association

data of the OAS1 splice-site SNP with viral clearance which includes WNV (Kim et al. 2009).

Minor issues:You reference a publication, but it would be useful to show the DNA sequence of the region surrounding rs10774671 showing the alternative splices that occur on the 2 alleles.

We agree with the reviewer that the schematic of the DNA sequence surrounding the rs10774671 splice site SNP would clarify the different splicing of OAS1 isoforms. This is now included in the schematic shown in Figure 1A.

L175 ar preferentially should be are preferentially.

This typo has been corrected.

Reviewer #1 (Significance (Required)):SignificanceThis is a conceptual advance in that a specific isoform of OAS1 that is genetically determined has anti-viral activity in vitro against several important viruses.The authors cite the one paper that examines circulating OAS1 and the genetic variant in SARS-Cov-2 https://www.nature.com/articles/s41591-021-01281-1.The interested audience would include those interested in immunogenetics and immunology.My expertise is in the area of the genetics of viral infections, and I do not have the expertise to assess the cell biology aspects of this paper.Referee Cross-commentingI agree that the comments of reviewers 2 and 3 should be addressed.Reviewer #2 (Evidence, reproducibility and clarity (Required)):Most positive-strand RNA viruses replicate within modified host organelles, shielding viral RNA from cytosolic RNA sensors (such as RIG-I). In this study, Soveg et al. propose a novel mechanism by which viral RNA may be sensed by the p46 isoform of the OAS1 protein. Five alternatively spliced isoforms of human OAS1 have been described, their specific roles have not been defined. Here, the authors provide evidence that the OAS1 p46 isoform is targeted to Golgi membranes via its C-terminal prenylation motif. Endomembrane targeting of OAS1 p46 resulted in enhanced detection of viral RNA as well as antiviral activity against different positive-strand RNA viruses that replicate within modified host organelles. The authors found that antiviral activity was mediated through activation of RNase L. In comparison, the OAS1 p42 isoform also showed some RNase L dependent anti-viral activity but localized to the cytosol and nucleus. Targeting of p42 to endomembranes by expressing the p46 CaaX prenylation motif did not increase p42 binding to viral RNA or antiviral activity, suggesting additional properties of the p46 C-terminus contribute to its antiviral activity. The authors provide support for their findings from human genetic data in which the splice site SNP mediating synthesis of the OAS1 p46 isoform was associated with COVID19 severity.The experimental work is extensive and was performed on cell lines (lentiviral transduction of OAS1 and ACE2 constructs, siRNA knockdown of OAS1, CRISPR/Cas9 knockout of OAS1 and RNASEL genes, transient or stable) and primary human PBMCs and fibroblasts. Prenylation (geranylgeranylation) of OAS1 isoforms was assessed using a click-it chemistry immunoprecipitation approach. Cellular localization of proteins was studied by confocal microscopy. Importance of the CaaX prenylation motif on the OAS1 p46 was shown by expressing modified OAS1 isoforms in cell lines expressing or lacking the C-terminal CaaX motif and assessing cellular localization and function. Standard in vitro virus infection and quantification assays were performed with a range of viruses. Genetic OAS1 SNP association studies were performed by genotyping a local cohort of 34 severe COVID-19 cases and 99 healthy controls and validated using a dataset from a larger European cohort (>10.000 datasets).

We appreciate the reviewer’s assessment of our study and thoughtful review to improve the clarity of the manuscript.

Specific comments:Confocal microscopy: It seems that quantifications were performed on as little as six individual cells (e.g. 1F)? Information on how these cells were chosen or how often experiments were repeated is not provided in the legend. Co-localizations are difficult to interpret from images shown. All confocal quantifications in the manuscript should display means of several individual experiments. Number of repeated individual experiments and quantified cells per experiment have to be stated in the legend.

Confocal microscopy quantifications shown in the manuscript are from one representative experiment. We now have provided specifics on the number of independent biological replicates for the microscopy images in the respective figure legends. Showing the number of cells analyzed provides the most transparency in the way these data are analyzed. This point is made by the reviewer’s question, as they raised a valid point that one of the cell type (Daudi cells; Figure 1F) had only a few cells quantified. This would not have been apparent if we had just plotted means from each experiment. In regards to Daudi cells, these are difficult to image as they are small and are hard to get them on the slides. Colocalization clarity issues is due to reduced resolution of the images in the manuscript, as we were constrained by file size limitations for the initial submission of our manuscript, but we will provide high resolution images for publication. We have also provided insets to show co-localization in the revised manuscript. We are also happy to send high resolution images to the journal so that they can transfer them to the reviewers.

Expression of OAS1 isoforms increased drastically upon IFN-b treatment (cell lines, primary cells, Figure 1). The authors should provide information if IFNb was added to virus infection experiments with (untransfected) cell lines or primary cells (e.g. Figure 2F-I) to induce OAS1 expression. Was infection in primary cells tested +/- IFN-b addition? Was IFN-b production from infected cells measured?

THP-1 cells transfected with isoform-specific siRNAs against OAS1 were not pre-treated with rIFNb before EMCV infection, as these cells induce a strong IFN response against viral infection. In THP-1 cells viral infection is sufficient to induce endogenous expression of OAS1 protein isoforms (Figure 2H-G). However, in the primary human fibroblasts (Figure 2H-I), EMCV infection alone was not sufficient to induce OAS1 protein levels to be detectable by Western blot (see Author response image 1). This is in line with previous studies have shown that fibroblasts are weak IFN producers when compared to sentinel cells, such as macrophages or pDCs. To circumvent this issue, we induced OAS1 by pre-treating cells with 25 U/mL rIFNb, a concentration at which the cells induce an antiviral state that would still support EMCV infection and replication. This is stated in the figure legend for Figure 2H-I of the manuscript.

**Author response image 1. sa2fig1:** Primary human fibroblasts do not induce OAS1 expression upon infection with EMCV. Primary human fibroblasts of (A/A) and (A/G) genotype at rs10774671 were infected with EMCV (MOI=0.01) or treated with 25 U/mL rIFNβ for 24h. OAS1 expression was assessed by Western blot.

Information on PBMCs and primary human fibroblasts seems missing (isolation and culture).

Information on PBMC and primary human fibroblast isolation and culture is now provided in the Methods section of our revised manuscript.

It is unclear from the information in the figure legends if experiments were adequately replicated. Figure legends should state how often experiments were repeated, if results from a representative or average of all experiments are displayed. Statistical methods seem adequate but should be calculated from all replicates of the experiments (which should be stated in the legend).

Number of replicates/independently performed experiments and whether data shown is representative or averaged is now stated in the respective figure legends.

Minor questions:Geranylgeranylation pull-down assay not explained in detail. Specific? Are there other methods?

To address this, we have now expanded the description of the click-chemistry labeling experiment in the Results section, and we describe the procedure in more detail in the Methods section. Click-chemistry labeling approaches are the gold standard for labeling and assessing small post-translational protein modifications for which suitable antibodies do not exist. We have modified the original protocol for the Click-iT labeling kit (Thermo) slightly by performing an OAS1 isoform-specific FLAG-IP prior to the click-chemistry labeling, which was performed ‘on-bead’. This facilitates a clear readout on whether the protein of interest is geranylgeranylated or not.

Western blots (Figure 1, S1): To allow visualization of all possible OAS1 isoforms, the range from 35 to 60 kDa should be shown on the images, ideally with a marker as reference (Western blot images range only from 3750 kDa). Lysates from the A549 cells ectopically expressing different OAS1 isoforms might have been used in WB to visualize the size of the different isoforms.

Based on the reviewer’s suggestion, we have extended the range on our primary cell Western blot (Figure 1C) and these data show that no other isoforms other than p42 and p46 are expressed in primary cells of (A/A), (A/G), or (G/G) genotype. Similar to the primary cells, none of the other OAS1 isoforms were detected by immunoblot in other cell-lines we tested.

Suggested improvements:The manuscript text contains text duplicates many places which impairs reading.

We thank the reviewer for noticing this issue caused due to formatting issues in MS Word. We have now fixed those duplicate sections.

Too much and too detailed (result text and figures), could be shortened (some data removed) and focused to emphasise the major findings.

We really appreciate this reviewer’s comment, especially when revision puts an enormous burden on the authors to generate new supportive data. Most of the data that are supportive to the story are in the supplement.

Major question for discussion or further experiments:The manuscript opens by saying replication within host organelles shields viral RNA from sensing e.g. by RIGI/MDA-5, and targeting of host viral sensors to these would be beneficial. However, since p46 is targeted to the cytosolic side of endomembranes – and supposedly co-localises with viral RNA – this suggests that viral replication is not within organelles, but rather located to the cytosolic side of endomembranes. How, then, is RNA shielded from RIG-I/MDA-5? Authors should discuss or rewrite this part, alternatively use more markers (other golgi/ER markers, Rabs…?) to identify the precise localisation better. In addition, imaging should be improved to show co-localisation better (high-resolution insets?) underlying quantifications. Purple and red are not ideal choices combined in images.

The RNA replication for positive-strand RNA viruses occurs within the replication organelle. Our confocal and viral RNA immunoprecipitation experiments suggest either p46 gains access to the replicating organelle where it can detect vRNA as it is replicating, or p46 is proximal to sites of vRNA export. We also

have provided high-resolution insets for the confocal microscopy panels in Figure 6B and S2C.

To address the reviewers’ question regarding the placement of the viral protein in relation to the replication organelle, Author response image 2 provides a schematic on how OAS1 could interact with the viral RNA. Based on our data, we propose that prenylated OAS1 p46 localizes to the cytosolic surface of the organelle membrane. During infection with a positive-strand RNA virus, the viral non-structural proteins and viral RNA are recruited to the organelle surface, where p46 can interact with vRNA. As the membrane is involuted to form the viral replication organelle (VRO), prenylated p46 is also included in the VRO. Confinement of p46 in the VRO may enhance its access to viral RNA. Furthermore, newly prenylated proteins are continuously recruited to the organelle surface during infection, which may allow for additional interference of vRNA as it escapes the replication organelle. We have now added this in the Discussion section of the manuscript.

Some specifics:Text duplicates: Lines 80-86 (same as 86-92), line 105-111 (same as 111-118), lines 170-176 (same as 176183). Some duplicate passages seem to have minor differences or lack references. Revise carefully.

We thank the reviewer for noticing this issue caused by formatting issues with Word. The duplicate sections are now removed from the revised manuscript.

Line 199: Until here, the manuscript did not show expression in primary human fibroblasts (Figure 1C shows PBMCs). OAS1 expression in fibroblasts is only shown in Figure S2C (images difficult to interpret, not quantified)?

Thank you for pointing to this error. This is a typo, we meant to say PBMCs in reference to Figure 1C, and not primary human fibroblasts. This has been corrected in the revised manuscript. For Figure S2C we have now included high-resolution insets to provide a clearer picture of OAS1/Golgin-97 co-localization.

Line 203: Check referenced figure numbers.

The correct Figures (1B, 1C, and S2C) are now referenced.

Lines 208-220 are difficult to read and need revision.

This section of the manuscript has now been revised for clarity.

The rationale for using Huh7 cells ectopically expressing p42 or p46 in Figure 1G/H should be explained (lines 215-217).

We chose Huh7 hepatoma cells specifically for our imaging analyses because these cells are easily transfectable and amenable to microscopy as they have a large cytosolic compartment which facilitates imaging/microscopy of cytosolic proteins. This information is included in the manuscript.

Figure 1D, G and H: Information should be added to the Figure, stating which cell type was analyzed.

For Figure 1D we annotated *OAS1* KO 293T below the Western blot. This information has now also been included in the respective figure legend.

For Figure 1G and H *OAS1* KO Huh7 cells were used. This is stated in the figure legends, and we have now also labeled Figure 1G-H accordingly.

Some of the Supplementary Figures could be re-arranged to correspond mainly to one main Figure: Figure S1 could be split (S1A+B, part C-M become new Figure). Figure S2C is more related to Figure 1 and S1.

Now, we have carefully revised the supplement to make sure the figures in the supplement match with

the text.

Table S1 important, but difficult to interpret. A summary of the data for the genotypes (not split by ancestry) or in a graph would help.

We have revised Table S1 and provided a graphical depiction of the data to the main figures (Figure 6G), based on the reviewer’s suggestions.

Reviewer #2 (Significance (Required)):The specific roles of the different human OAS1 isoforms in antiviral immunity have not yet been identified. The study provides a conceptual advance by demonstrating that the OAS1 p46 isoform, via prenylation of a Cterminal CaaX motif, is targeted to Golgi membranes. By adding or removing the prenylation motif to different OAS1 isoforms as well as further modifications, the authors could narrow down the effect of the p46 isoform mainly, but not exclusively to the C-terminal prenylation motif and found that the antiviral activity was mediated via RNaseL. It is further of interest that the p46 isoform also had antiviral activity against SARS CoV-2 and that alleles encoding for the cytosolic isoform OAS1 p42 were overrepresented in seriously ill COVID-19 patients (two cohorts).This is the first study to identify a function specific to one of the OAS p46 isoform. Targeting of the OAS1 p46 RNA sensor isoform to endomembranes might present a novel host defense mechanism to detect RNA from viruses replicating in modified host organelles.The study demonstrates a general antiviral mechanism and findings should be of interest to a broad audience (especially with the increased focus on viral infections due to the SARS CoV-2 pandemic). The study could direct increased focus on differential functions of isoforms of immune proteins.Our expertise: Immunology, host-pathogen interactions, host defense to infectionReferee Cross-commentingI agree with the comments from the other reviewers.Reviewer #3 (Evidence, reproducibility and clarity (Required)):The work by Soveg and Schwerk et al. presents findings on the role of an isoform of OAS1, p46, on control of viral infection. This work concludes that the p46 isoform targets to endomembranous compartments of the cell, which are replication sites for some viruses. In the case of infection with these viruses, p46 confers protection to cells against virus infection, by targeting the antiviral p46 to the appropriate sites of viral replication. The authors demonstrated this by a series of biochemical and microscopy assays, mostly on protein overexpression systems. They complemented the work with experiments on endogenous OAS1, and human genetic studies linking the p46 isoform genetic variant with susceptibility to COVID-19. Overall, this was a well executed study showing clearly that the targeting of OAS1 to the site of virus infection is important for its antiviral activity.The conclusion that p46, with is distinct subcellular localization, confers better protection against viral infection compared to p42 is very convincing. The molecular manipulations of p46 C-terminal domains demonstrate the importance of the CaaX domain as well as the length of the C-terminal region. However, one of the major weaknesses of this study is the reliance on overexpression of the OAS1 proteins rather than studying endogenous proteins in the context of viral infection. In some experiments, the authors addressed this by investigating endogenous isoforms in THP1 and primary human fibroblasts. In these experiments, the role of OAS1 on control of EMCV is overall less strong, but there is nevertheless a difference between p42 and p46 isoforms that is clear.

We appreciate this reviewer’s insights and suggestions. We have addressed your concerns in our point-by-point response below:

Additional improvements to the manuscript are as follows:Major comments:The major outcomes of viral infection measured in the study were viral titres and RNA levels. Could the authors employ additional experiments to look at other outcomes? E.g. measurement of cellular 2'-5'-A produced, cell death, or cellular RNA degradation (i.e. RNaseL activation)?

While we agree with the reviewer that measuring another parameter could be helpful, there are no assays that are sensitive enough that can provide quantitative measurement of endogenous 2’-5’A production in infected cells. In the initial phase of our study, we also tried to assess RNase L activation by measuring RNA degradation using bioanalyzer. Although the OAS/RNase L community have shown laddering mostly in A549 cells, we did not observe RNA laddering as a readout for RNase L activation in a variety of cells tested. As we do not observe host RNA degradation/laddering, this might indicate a more localized viral RNA-specific degradation induced by RNaseL. We show that OAS1 isoforms produce 2’-5’A and its antiviral activity is dependent on intact RNase L (see Figures 3D-F), demonstrating that this pathway is required for the antiviral activity of p46. Based on these issues, quantification of the viral titer by plaque assay is most sensitive and biologically relevant readout when assessing the activity of OAS1 isoforms.

As the authors mentioned in the discussion, often there are examples of viral antagonism in important innate immune pathways. The authors briefly touch on this. Could they provide more detailed examples of OAS1 targeting specifically by viruses relevant to sensing at the ER/Golgi membranes? This will help demonstrate the important role of OAS1 p46 in innate immunity to virus infection.

While a few viruses are known to antagonize OAS-RNAseL pathway, among positive sense RNA viruses Mouse Hepatitis Virus (MHV) and MERS-CoV accessory protein encodes for a phosphodiesterase which degrade 2’-5’A (PMIDs: 27025250, 19176619). As 2’-5’A is critical for the activation of RNAseL, antagonizing this pathway makes them insensitive to OAS1-RNaseL mediated virus control. However, intriguingly, SARSCoV-2 do not carry this phosphodiesterase in their genome, making them susceptible to OAS1-p46. This is now included in the discussion.

The link found between OAS1 p42/p46 isoforms and COVID-19 lends weight to the importance of this protein in innate immunity. However, the authors only compared between severe cases and uninfected individuals. Would a more appropriate comparison be between severe and mild/asymptomatic cases?Furthermore, graphical depiction of this data could potentially be useful for the reader to appreciate the effects.It would also be useful if the authors more directly described the prevalence of rs10774671 A/G variants in the general population.

Unfortunately, we do not have enough patients with mild/asymptomatic COVID-19 to give us sufficient power for such analysis. None of the recently published genetic studies including our replication cohort have recruited mild/asymptomatic COVID-19 patients (Pairo-Castineira et al., Nature 2021). However, the reviewer has a valid point, studies on mild/asymptomatic should be carried out when we are out of the pandemic phase since current priority has been to treat severe covid-19 patients.

Based on the reviewers advice, we have now included a graphical depiction of our genetic association data in the main figure (Figure 6G) as requested. We have also included a graphical depiction of the prevalence of rs10774671 in the global human population (Figure S1B).

Minor comments:There are several instances of text duplication throughout the manuscript – lines 86-92, 176-183, 618-627, 760-761.

We thank the reviewer for noticing this issue caused by formatting issues. We have now fixed those duplicated sections.

The geranylgeranyl click chemistry experiment could be better explained in the text, when it is presented in the Results section.

Based on the reviewer’s comment, we have expanded the description of the click chemistry experiment in the Results section.

Reviewer #3 (Significance (Required)):This work demonstrates am important advancement in our understanding of OAS1 function, and its potential relevance during viral infection. The context of genetic studies linked to COVID-19 highlights the potential importance of this protein in human health.This being said, this work overall is incremental in the context of the role of OAS1 in antiviral innate immunity, although it demonstrates the importance of subcellular localisation for the function of nucleic acid-sensing innate immune receptors. This is something that has not been investigated in much depth, and continues to be not well understood for many other receptors in the field. This work opens the possibility that subcellular localization of other receptors, or their isoforms, could similarly have effects on their function, and that investigations of localization will be important avenues of research in the field in future. This will certainly be of interest to the field of viral immunology, and nucleic acid sensing in particular (constituting my area of expertise).Referee Cross-commentingI agree with the comments from the other reviewers.